# MEST: Accurate and Fast Memory-Economic Sparse Training Framework on the Edge

**Geng Yuan**[1,†]**, Xiaolong Ma**[1,†]**, Wei Niu**[2]**, Zhengang Li**[1]**, Zhenglun Kong**[1]**, Ning Liu**[3]**,**
**Yifan Gong**[1]**, Zheng Zhan**[1]**, Chaoyang He**[4]**, Qing Jin**[1]**, Siyue Wang**[1]**,**
**Minghai Qin, Bin Ren**[2]**, Yanzhi Wang**[1]**, Sijia Liu**[5]**, Xue Lin**[1]

[1] Northeastern University, [2] College of William and Mary,
[3] Midea Group, [4] University of Southern California, [5] Michigan State University
{yuan.geng,xue.lin}@northeastern.edu

## Abstract

Recently, a new trend of exploring sparsity for accelerating neural network training has emerged, embracing the paradigm of training on the edge. This paper proposes a novel Memory-Economic Sparse Training (MEST) framework targeting for accurate and fast execution on edge devices. The proposed MEST framework consists of enhancements by Elastic Mutation (EM) and Soft Memory Bound (&S) that ensure superior accuracy at high sparsity ratios. Different from the existing works for sparse training, this current work reveals the importance of *sparsity schemes* on the performance of sparse training in terms of accuracy as well as training speed on real edge devices. On top of that, the paper proposes to employ data efficiency for further acceleration of sparse training. Our results suggest that unforgettable examples can be identified *in-situ* even during the dynamic exploration of sparsity masks in the sparse training process, and therefore can be removed for further training speedup on edge devices. Comparing with state-of-the-art (SOTA) works on accuracy, our MEST increases Top-1 accuracy significantly on ImageNet when using the same unstructured sparsity scheme. Systematical evaluation on accuracy, training speed, and memory footprint are conducted, where the proposed MEST framework consistently outperforms representative SOTA works. Our codes are publicly available at: `https://github.com/boone891214/MEST`.

## 1 Introduction

To promote the broader applications of deep learning on the edge, a surge of research efforts have been devoted to removing the over-parameterization in neural networks for accelerated inference. Specifically, existing works have explored various strategies such as heuristics-based pruning [1, 2], regularization-based pruning [3, 4], and recently prevailing network architecture search [5, 6, 7, 8].

Recently, a new trend of exploring sparsity for training acceleration of neural networks has emerged to embrace the promising training-on-the-edge paradigm. The first works in this direction use the pruning-at-initialization approach such as SNIP [9] and GraSP [10] that first obtains a fixed sparse model structure and then follows with a traditional training process. However, the whole process is still computation- and memory-intensive, and therefore not compatible with the end-to-end edge training paradigm. Such a sparse training methodology with the pre-fixed structure also faces the problem of compromised accuracy.

---

† These authors contributed equally.

35th Conference on Neural Information Processing Systems (NeurIPS 2021).

Furthermore, sparse training with dynamic sparsity mask such as SET [14], DeepR [15], DSR [13], RigL [11], and SNFS [12] have been proposed, showing great potential towards end-to-end edge training. Specifically, they start with a sparse model structure picked intuitively from the initialized (but not trained) dense model, and then heuristically explore various sparse topologies at the given sparsity ratio, together with the sparse model training. The underlying principle of sparse training is that the total epoch number is the same as dense training, but the speed of each training iteration (batch) is significantly improved, thanks to the reduced computation amount due to sparsity.

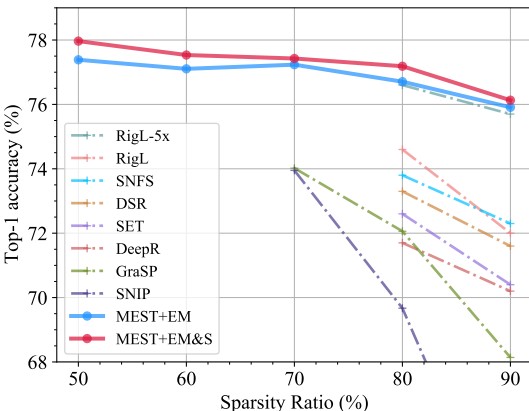

Figure 1: Accuracy vs sparsity ratio on ImageNet using ResNet-50 dense model. Our proposed MEST framework: MEST+EM (Elastic Mutation) and MEST+EM&S (with Soft Memory Bound) are compared with the SOTA sparse training algorithms i.e., GraSP [10], SNIP [9], RigL [11], SNFS [12], DSR [13], SET [14], and DeepR [15].

This paper proposes a novel Memory-Economic Sparse Training (MEST) framework targeting for accurate and fast execution on edge devices. Specifically, we boost the accuracy with the MEST+EM (Elastic Mutation) to effectively explore various sparse topologies. And we propose another enhancement through applying a soft memory bound, namely, MEST+EM&S, which relaxes on the memory footprint during sparse training with the target sparsity ratio met by the end of training. In Figure 1, the accuracy of the proposed MEST is compared against state-of-the-art (SOTA) sparse training algorithms under different sparsity ratios, using the same unstructured sparsity scheme.

Furthermore, as with inference acceleration, we find that sparse training closely relates to the adopted sparsity scheme such as unstructured [16], structured [17, 18], or fine-grained structured [19] scheme, which can result in varying accuracy, training speed, and memory footprint performance for sparse training. With our effective MEST framework, this paper systematically investigates the sparse training problem with respect to the sparsity schemes. Specifically, besides the directly observable model accuracy, we conduct thorough analysis on the memory footprint by different sparsity schemes during sparse training, and in addition, we measure the training speed performance under various schemes on the mobile edge device.

On top of that, the paper proposes to employ data efficiency for further acceleration of sparse training. The prior works [20, 21, 22, 23] show that the amount of information provided by each training example is different, and the hardness of having an example correctly learned is also different. Some training examples can be learned correctly early in the training stage and will never be "forgotten" (i.e., misclassified) again. And removing those easy and less informative examples from the training dataset will not cause accuracy degradation on the final model. However, the research of connecting training data efficiency to a sparse training scene is still missing, due to the dynamic sparsity mask. In this work, we explore the impact of model sparsity, sparsity schemes, and sparse training algorithm on the amount of removable training examples. And we also propose a data-efficient two-phase sparse training approach to effectively accelerate sparse training by removing less informative examples during the training process without harming the final accuracy. The contributions of this work are summarized as follows:

- A novel Memory-Economic Sparse Training (MEST) framework with enhancements by Elastic Mutation (EM) and the soft memory bound targeting for accurate and fast execution on the edge.

- A systematic investigation of the impact of sparsity scheme on the accuracy, memory footprint, as well as training speed with real edge devices, providing guidance for future edge training paradigm.

- Exploring the training data efficiency in the sparse training scenario for further training acceleration, by proposing a two-phase sparse training approach for *in-situ* identification and removal of less informative examples during the training without hurting the final accuracy.

- On CIFAR-100 with ResNet-32, comparing with representative SOTA sparse training works, i.e., SNIP, GraSP, SET, and DSR, our MEST increases accuracy by 1.91%, 1.54%, 1.14%, and 1.17%;

achieves $1.76\times$, $1.65\times$, $1.87\times$, and $1.98\times$ training acceleration rate; and reduces the memory footprint by $8.4\times$, $8.4\times$, $1.2\times$, and $1.2\times$, respectively.

## 2 Background and Related Work

This section introduces representative neural network sparsity schemes and their impacts on memory footprint in sparse training, as well as the neural network sparse training strategies.

### 2.1 Sparsity Scheme

Neural network pruning has been well investigated for inference acceleration. The majority of works in this direction apply a pretraining-pruning-retraining flow, which is not compatible with the training-on-the-edge paradigm. According to the adopted sparsity scheme, those works can be categorized as *unstructured* [16, 1], *structured* [24, 2, 25, 26, 17, 3, 27, 28, 29, 30, 31, 18, 32, 33], and *fine-grained structured* [19, 34, 35, 36, 37, 38, 39, 40, 41] including the pattern-based and block-based ones. Detailed discussion about these sparsity schemes is provided in Appendix A.

Although these sparsity schemes are mainly proposed for accelerating inference, we find that they also play an important role in sparse training in terms of accuracy, memory footprint, and training speed. For memory footprint, we focus on the two major components that vary with the sparsity scheme: *model representation* together with *gradients* produced during training. The detailed analysis is provided in Appendix B.

### 2.2 Sparse Training

Majority of the sparse training works can be categorized into two groups: fixed-mask sparse training and dynamic-mask sparse training. Additionally, there exist works [42, 43, 44] that prune dense networks in the early training stage, but they are out of scope for sparse training on the edge.

#### 2.2.1 Sparse Training with Fixed Sparsity Mask

The fixed-mask approach [9, 10, 45, 46, 47] has been proposed to decouple pruning and training such that after pruning, the sparse model training can be executed on edge devices. SNIP [9] preserves the loss after pruning based on connection sensitivity. GraSP [10] prunes connections in a way that accounts for their role in the network's gradient flow. SynFlow [45] proposes iterative synaptic flow pruning, which avoids layer collapse and preserves the total flow of synaptic strengths throughout the network. Since the proposed pruning algorithm does not incorporate back propagation, it achieves global pruning at initialization without data. Based on the unstructured SNIP objective, 3SP [47] further introduces a computation-aware weighting of the pruning score. This actively biases pruning by removing more computation-intensive channels which either have small effect on the loss or have significant computation cost. However, the pruning-at-initialization process is still computation and memory-intensive and therefore not compatible with the end-to-end sparse training on the edge. And these works (except 3SP) employ the unstructured sparsity scheme.

#### 2.2.2 Sparse Training with Dynamic Sparsity Mask

To reduce the computation as well as memory footprint during the whole training phase, sparse training is exploited in many works [15, 14, 13, 12, 11], which can adjust the sparsity topology during training as well as maintain a low memory footprint. Sparse Evolutionary Training (SET) [14] uses magnitude-based pruning and random growth at the end of each training epoch. DeepR [15] combines dynamic sparse parameterization with stochastic parameter updates for training. This method is primarily demonstrated with sparsification of fully-connected layers and applied to relatively small and shallow networks. DSR [13] develops a dynamic reparameterization method to achieve high parameter efficiency in training sparse deep residual networks. SNFS [12] develops sparse momentum, an algorithm which uses exponentially smoothed gradients (momentum) to identify layers and weights which reduce the error efficiently. RigL [11] proposes to iteratively update sparse model topology during training by calculating dense gradients only at the update step. Note that, though SNFS and RigL are sparse training, they actually involve computation of all the gradients corresponding to both pruned and non-zero weights, and therefore their memory footprint is equivalent to that of dense training.

# 3 Sparse Training on the Edge

## 3.1 Sparsity Scheme in Sparse Training on the Edge

It is common to see that sparse training works [9, 10, 15, 14, 13, 12, 11, 45] represent the training speed performance using the training FLOP count. Actually, such FLOPS cannot account for the actual execution overheads caused by the sparse data operations. For example, the unstructured sparsity exhibits an irregular memory access pattern, leading to significant execution overhead. Moreover, the dense model can take advantage of Winograd [48] to significantly accelerate the computation speed, which may apply in sparse computation. Therefore, this paper directly measures the training speed performance on a mobile device. We investigate the training acceleration performance by different sparsity schemes including unstructured, structured, and two state-of-the-art fine-grained structured (i.e., block and pattern), through a prototype implementation on a mobile edge device. To do so, we conduct compiler-level optimizations, leveraging a computation-graph based compilation approach, including optimizations on computation graph itself, tensor optimizations, etc. More details are provided in Appendix C.

Figure 2 shows representative training speed performance under various sparsity schemes, using an example CONV layer adopted from ResNet-32. We evaluate the acceleration rate by measuring the total forward and backward execution time of the sparse CONV layer on a Samsung Galaxy S20 smartphone, and then normalizing with respect to that of the corresponding dense layer. Surprisingly, even under the same sparsity ratio, we found that the acceleration rates of different sparsity schemes are varied significantly. When the sparsity ratio is below 70% and 80% for block-based sparsity and unstructured sparsity schemes, respectively, they cannot achieve any acceleration and even slow down the computation speed, compared to the corresponding dense model. Thus, choosing an appropriate sparsity scheme is an essential factor for sparse training for acceleration purposes.

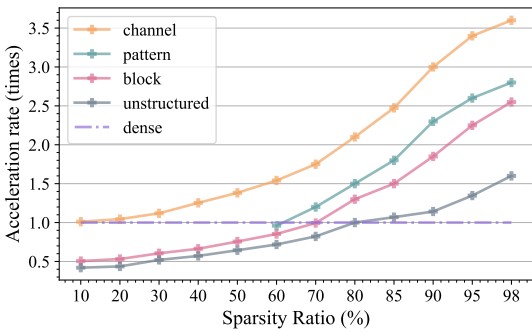

Figure 2: Training acceleration rate vs sparsity ratio of different sparsity schemes. The results here are measured on a $3\times3$ CONV layer selected from ResNet-32 with 64/64 input/output channels and $16\times16$ input feature map, with a Samsung Galaxy S20 smartphone.

## 3.2 Proposed Memory-Economic Sparse Training (MEST) Framework

To facilitate the sparse training on edge devices, our MEST framework is designed for the following objectives: 1) towards end-to-end memory-economic training by considering the resource limitation of edge devices; 2) Exploiting sparsity schemes to achieve high sparse training acceleration while maintaining high accuracy. We propose the MEST method (vanilla) to periodically remove less important non-zero weights from the sparse model and grow zero weights back during the sparse training process, which we call **mutation**, to explore desired sparse topologies with a specified *sparsity scheme* and *ratio*. Previous works [14, 13] directly use the weight magnitude as the indicator of its importance. However, determining the weight importance only based on its magnitude is not ideal, because the weight magnitude may fluctuate significantly during the training. Therefore, in MEST, we incorporate the weight's current gradient as an indicator for its changing trend to estimate its importance. We define the importance score as:

$$Scr_{w_\tau} = |w_\tau| + |\lambda \frac{\partial \ell(W_{\tau-1}, D)}{\partial w_{\tau-1}}|, \tag{1}$$

where $D$ denotes the training data; $\ell(W_{\tau-1}, D)$ and $\frac{\partial \ell(W_{\tau-1}, D)}{\partial w_{\tau-1}}$ are the loss and gradient at epoch $\tau$; and $\lambda$ is the coefficient for the gradient. As a result, three types of weights are considered relatively important, which are the weights with 1) large weight magnitude but small gradient, 2) small weight magnitude but large gradient, and 3) large weight magnitude and large gradient. The exploration of the impact of the coefficient $\lambda$ on sparse training accuracy is shown in Appendix E.

More importantly, different from the methods (e.g., RigL [11]) that use gradients of the dense model to find the weights to grow back, we only use sparse gradients to identify less important

weights to remove, then randomly select weights to grow back. In this way, our MEST strictly keeps the sparsity of weights and gradients during the entire sparse training process. This is critical for memory-economic sparse training on resource-limited edge devices. Moreover, different from previous dynamic sparse training methods that only uses unstructured sparsity, our MEST consider the constraints of different sparsity schemes into the mutation policy (more details in Appendix D).

**Elastic Mutation (EM):** We further propose an Elastic Mutation method to gradually reduce the mutation rate along with the training process, called Memory-Economic Sparse Training with Elastic Mutation (MEST+EM). We are mainly based on two considerations: 1) a larger mutation ratio will provide a larger search space during the dynamic sparse training process; and 2) the dramatic structural change of the network may compromise the training convergence. Thus, we propose our EM method to gradually reduce the mutation ratio during the dynamic sparse training process, which maintains a sufficient search space while making the sparse model smoothly stabilize to an optimized structure.

***Soft memory bound* Elastic Mutation (EM&S):** If the application scenario that the memory footprint could be a soft constraint, we propose an enhancement with the *Soft Memory Bound*, namely, Memory-Economic Sparse Training with *Soft-bounded* Elastic Mutation (MEST+EM&S), as

---

**Algorithm 1:** MEST with (Soft) Elastic Mutation

**Input:** Network with uninitialized weight $W$ in a total of $L$ layers, target sparsity ratio $s, p, \tau, \Delta\tau, \tau_{stop}$.
**Output:** A sparse model satisfying the target sparsity requirement.
Initialize $W$ with random values and random sparse mask according to the sparsity requirements.
**while** $\tau < \tau_{stop}$ **do**
  **if** *MEST+EM* **then**
    **if** $(\tau \mod \Delta\tau) = 0$     ▷ do weight mutation
    **then**
      Decay $p$ if $\tau$ reaches a decaying milestone.
      **for** *each layer weight tensor $W^l$* **do**
        $W^l \leftarrow \text{ArgRemoveTo}(W^l, s+p)$
        $W^l \leftarrow \text{ArgGrowTo}(W^l, s)$

  **if** *MEST+EM&S* **then**
    Decay $p$ if $\tau$ reaches a decaying milestone.
    **for** *each layer weight tensor $W^l$* **do**
      $W^l \leftarrow \text{ArgGrowTo}(W^l, s-p)$
    Training for $\Delta\tau$ epochs;      ▷ $\tau \leftarrow \tau + \Delta\tau$
    **for** *each layer weight tensor $W^l$* **do**
      $W^l \leftarrow \text{ArgRemoveTo}(W^l, s)$

Continue sparse training from the epoch $\tau_{stop}$ to $\tau_{end}$.

---

an option to further improve accuracy. Different from the EM method that the less important weights will always be removed, our EM&S allows the newly grown weights to be added to the existing weights and then trained, then the less important weights will be selected from all weights including the newly grown weights. This can avoid forcing the existing weights in the model to be removed if they are more important than newly grown weights. This can be considered as adding an 'undo' mechanism to the mutation process. Note that even with a soft memory bound, the target sparsity ratio can still be met by the end of sparse training and keep the entire training process sparse.

***Notation and Preliminary:*** Consider $W \in \mathbb{R}^N$ is the weights of the entire network. The number of weights in the $l$-th layer $W^l$ is $N^l$. Our target sparsity ratio is denoted by $s \in (0, 1)$. We mutate on a fraction $p \in (0, 1)$ of the weights in $W^l$. Suppose the total number of training epoch is $\tau_{end}$, then we conduct the weight mutation for the first $\tau_{stop}(< \tau_{end})$ epochs with a frequency of $\Delta\tau$.

Algorithm 1 shows the flow of MEST+EM and MEST+EM&S. The main difference between MEST+EM and MEST+EM&S is the order that $\text{ArgRemoveTo}(\cdot)$ and $\text{ArgGrowTo}(\cdot)$ are performed. In MEST+EM, we perform weight mutation for every $\Delta\tau$ epochs with following steps: first use $\text{ArgRemoveTo}(W^l, s+p)$ to remove $p \times N^l$ less important weights from a total of $s \times N^l$ non-zero weights; and then grow the model back to sparsity $s$ with $\text{ArgGrowTo}(W^l, s)$, which randomly activates a number $p \times N^l$ of zero weights. The newly activated weights although are being zeros, will be considered as part of the sparse model and be trained. During the entire training process, the model sparsity is strictly bounded by $s$, thus maintaining a hard memory bound. On the other hand, in MEST+EM&S, for every other $\Delta\tau$ epochs, we first grow the model to reduce the sparsity ratio to $s - p$ by $\text{ArgGrowTo}(W^l, s-p)$ and train for $\Delta\tau$ epochs, and then remove weights to increase the sparsity ratio back to $s$ by $\text{ArgRemoveTo}(W^l, s)$. We also decay $p$ at given milestone epochs until $\tau_{stop}$. During the entire training process, the weights are trained at sparsity ratio $s - p$, and sparsity

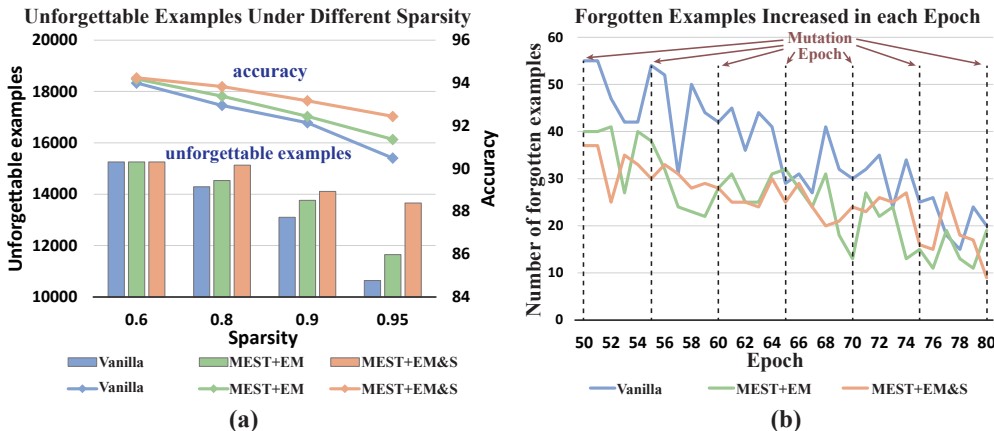

Figure 3: Data efficiency investigation on ResNet-32 using CIFAR-10. (a) The number of unforgettable examples after the sparse training process under different sparse training algorithms and sparsity ratios; (b) The number of increased forgotten examples in each epoch (between epoch 50 to 80 with a mutation frequency of 5 epochs).

is gradually increasing to the target sparsity ratio through the decay of $p$. In addition, the mutation process is actually operated on indices, to facilitate implementation on the edge device.

## 4  Exploring Data Efficiency in Sparse Training

Data efficiency has been studied for the traditional training in literature [20, 21, 22, 23]. It has been proven that the amount of information provided by each training example to a network is different, and the difficulty of learning examples also varies. Some training examples are easily learned at early training stage. And once some examples are learned, they will never be "forgotten" (i.e., misclassified) again. Removing those easy and less informative examples from the training dataset will not cause accuracy degradation on the final model. More details is discussed in Appendix F. However, the exploration of data efficiency in sparse training scenarios is still missing. Due to the dynamic sparsity mask generation in sparse training, it is still unknown that whether the data efficiency can be leveraged for further accelerating sparse training. Therefore, we need to first figure out the impact of model sparsity on the number of removable training examples (e.g., unforgettable examples), and then discuss the possibility of leveraging data efficiency for accelerating sparse training.

### 4.1  Impact of Model Sparsity on Dataset Efficiency

In [23], they proposed to use the number of forgetting events of a training example along the entire training process to indicate the amount of information of the example. The forgetting event is defined as an individual training example transitions from being classified correctly to incorrectly over the training process. It can also be considered as the example is forgotten by the network. An example can be forgotten multiple times. An unforgettable example stands for an example that has never been forgotten once it is correctly classified, and it is considered less informative to a network and easy to be learned [23]. More details is shown in Appendix F.

In order to study whether data efficiency can be used to accelerate sparse training, we first explore the number of unforgettable examples that can be identified after the sparse training process under different sparsity ratios. We test with our three sparse training algorithms MEST (vanilla), MEST+EM, and MEST+EM&S. The Figure 3 (a) shows the results obtained on ResNet-32 using CIFAR-10 dataset. We find that there is still a considerable portion (30%∼34%) of training examples in CIFAR-10 dataset are unforgettable to a highly sparse network (under 95% sparsity). The number of unforgettable examples decreases as the model sparsity increases, and it shows a positive correlation with the model accuracy. When under a high sparsity, the network generalization performance decreases, making some easy examples harder to remember. Moreover, we also observe that a better sparse training algorithm (e.g., MEST+EM&S) leads to more unforgettable training examples, indicating the potential of removing a larger portion of training examples and hence a higher acceleration.

## 4.2 Will Mutation Lead to Forgetting?

It is a natural question to ask that whether the structure change during the training such as our proposed Elastic Mutation will lead to severe forgetting. Thus, we evaluate the number of unforgetable examples in the epoch before and after the mutation. Figure 3 (b), shows the number of forgotten examples increased in each training epoch, which equals the difference of unforgettable examples between two consecutive epochs. Neither the mutation in MEST+EM nor MEST+EM&S causes a notable increase in forgetting. This is because the mutated weights are least important, which have a minor impact on the model accuracy. Detailed results are shown in Appendix F.

## 4.3 Data-Efficient Sparse Training on the Edge

To identify the less informative training examples, prior work [23] collects the statistics of forgetting events through the entire training process. Then, using compressed dataset to train the network from scratch. Obviously, this is not an efficient, even an unaffordable solution for training on edge devices.

Different from prior works, we intend to integrate the less informative example identification and data-efficient sparse training into one single training process. Our objective is to obtain a similar final accuracy as a full dataset training within the same number of training epochs. Thus, we propose a data-efficient training method (DE), which separate one training process into two phases. For the first training phase, the full dataset is used for a certain sparse training epochs while counting the number of forgetting events for each example. The first phase takes several epochs (e.g., 70) to obtain a stable identification results. For the second training phase, partial training examples will be removed from the training dataset and obtain a compressed training dataset for the rest of the training process. The number of removed examples depends on the number of examples within a forgetting events threshold. Note that the examples that only be forgotten few times (e.g., 1 or 2) may also relatively easy to learn, which may also be removed without harming the accuracy. Denoting the full training dataset as $D$, the compressed training dataset $\hat{D}$ is described as:

$$\hat{D} = \{x_i | x_i \in D \text{ and } f(x_i) \leq th\}, \tag{2}$$

where the $x_i$ and $f(x_i)$ represent the $i$-$th$ training example in the full training dataset and its number of forgetting events occurred in the first training phase, and $th$ is a given threshold.

The Figure 4 shows the status of final accuracy obtained by the two-phase training approach under different sparsity ratios, sparsity schemes, our proposed mutation methods, number of forgotten thresholds, and epochs for the first phase training. The yellow grids mean that using that number of epoch for the first training phase can achieve similar accuracy as using a full dataset for the entire training process.

We have the following observations: 1) Compared to dense model, the sparse models takes longer to identify a good set of removable (less informative and easy) examples; 2) The larger number of examples to be removed, the more training epochs are required for the first training phase. 3) Unstructured sparsity scheme requires fewer epochs than fine-grained sparsity schemes (block and pattern). 4) MEST+EM and MEST+EM&S require similar number of epochs. 5) Besides the unforgettable examples, examples with few forgotten times are also removable without harming the final accuracy. More results on other dataset and networks can be found in Appendix F.

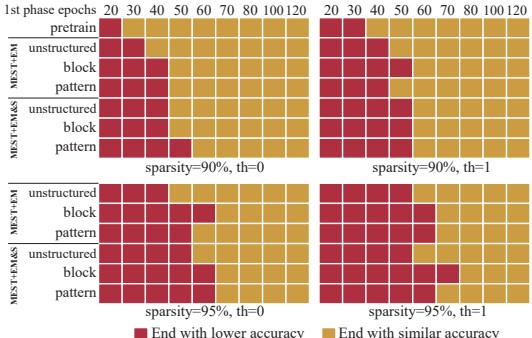

Figure 4: The epoch number used for the first training phase and its corresponding final accuracy status under sparsity ratios of 90% and 95% and threshold of 0 and 1. Results are obtained on ResNet-32 using CIFAR-10.

## 5 Experimental Results

This section evaluates the proposed MEST framework. The training speed results are obtained using a Samsung Galaxy S20 smartphone with Qualcomm Adreno 650 mobile GPU. We measure the

Table 1: Accuracy comparison with SOTA works using ResNet-32 on CIFAR-10 and CIFAR-100.

| Dataset | Memory Footprint | CIFAR-10 (*Dense: 94.88*) | | | CIFAR-100 (*Dense: 74.94*) | | |
|---|---|---|---|---|---|---|---|
| Sparsity ratio | | 90% | 95% | 98% | 90% | 95% | 98% |
| LT [49] | dense | 92.31 | 91.06 | 88.78 | 68.99 | 65.02 | 57.37 |
| SNIP [9] | dense | 92.59 | 91.01 | 87.51 | 68.89 | 65.22 | 54.81 |
| GraSP [10] | dense | 92.38 | 91.39 | 88.81 | 69.24 | 66.50 | 58.43 |
| DeepR [15] | sparse | 91.62 | 89.84 | 86.45 | 66.78 | 63.90 | 58.47 |
| SET [14] | sparse | 92.30 | 90.76 | 88.29 | 69.66 | 67.41 | 62.25 |
| DSR [13] | sparse | 92.97 | 91.61 | 88.46 | 69.63 | 68.20 | 61.24 |
| MEST (vanilla) | sparse | 92.12±0.13 | 90.86±0.11 | 88.78±0.26 | 69.35±0.36 | 67.85±0.23 | 62.58±0.31 |
| MEST+EM | sparse | 92.56±0.07 | 91.15±0.29 | **89.22±0.11** | **70.44±0.26** | **68.43±0.32** | **64.59±0.27** |
| MEST+EM&S | sparse | **93.27±0.14** | **92.44±0.13** | **90.51±0.11** | **71.30±0.31** | **70.36±0.05** | **67.16±0.25** |

computation time of a round of forward- and backward-propagation on a batch of 64 images to denote the training speed. The acceleration rate is the training speed of sparse training normalized to that of dense training. For accuracy, we repeat each experiment 3 times and report the mean and standard deviation of the accuracy results on CIFAR-10/100. For training speed, we report the average value from 100 runs. We use ResNet-32 and VGG-19 for CIFAR-10 and CIFAR-100 dataset [50], and ResNet-34 and ResNet-50 [51] for ImageNet-2012 [52]. Since the ImageNet-2012 is not practical for edge training, we mainly use it for accuracy (detailed explanations are in Appendix H). When combining our data-efficient two-phase training method that compresses the training dataset on the second phase, we denote our methods as MEST+EM+DE and MEST+EM&S+DE.

**Experimental setups:** We use the same training epochs as GraSP [10], which is $\tau_{end} = 160$ for CIFAR-10/100 and $\tau_{end} = 150$ for ImageNet. We use standard data augmentation, and cosine annealing learning rate schedule is used with SGD optimizer. For CIFAR, we use a batch size of 64 and set the initial learning rate to 0.1. For ImageNet, we use a batch size of 2048. Our learning rate is scheduled with a linear warm-up for 8 epochs before reaching the initial learning rate value of 2.048. We adopt a uniform sparsity ratio across all the CONV layers while keeping the first layer dense. The other reference works (except SET [14] and RigL [11] that use uniform sparsity) use non-uniform sparsity, which leads to a higher computation FLOPs compared to the uniform sparsity under the same sparsity ratio. An ablation study of using hybrid sparsity schemes and non-uniform sparsity ratio on MEST is shown in Appendix K. The hyper-parameter setting for elastic mutation are provided in Appendix G.

## 5.1 Accuracy Results

**CIFAR-10 and CIFAR-100:** The MEST accuracy results are shown in Table 1. We include the results at sparsity ratios of 90%, 95%, and 98% with unstructured sparsity scheme. Methods that use dynamic sparse training (DeepR, SET, and DSR) achieve slightly better results compared to fixed-mask sparse training. Compared to MEST (vanilla), which uses a fixed mutation ratio along with the training process, our MEST+EM consistently achieves higher accuracy. This proves the effectiveness and importance of our elastic mutation method in sparse training. And our MEST+EM&S further improves the accuracy significantly, especially in extremely high sparsity ratio (e.g. 98%). In terms of peack memory footprint, Lottery Ticket (LT), SNIP, and GraSP are equivalent to dense training. Because the SNIP and GraSP require computing the forward and backward propagation of a dense model to find a desired sparse structure. The LT method requires an iterative magnitude pruning process to find the "winning ticket" sparse structure first, it is also considered the same as the dense model in an end-to-end training scenario [49, 53, 54]. The VGG-19 results are in Appendix H.

**ImageNet-2012:** Table 2 shows the accuracy results and training FLOPS using ResNet-50. RigL [11] is a recent milestone of dynamic sparse training works, which has considerable improvements compared to previous works. To make a fair comparison with RigL, we scale our training epochs to have the same or less overall training FLOPs as the RigL. We also increase the training effort for MEST by 1.7×, which is 250 epochs, to compare with the RigL with 5× longer training, which is 500 epochs as reported in [11]. With the same or less training FLOPS, our proposed MEST framework consistently outperforms other baselines. When using our data effective training method,

Table 2: Accuracy comparison using ResNet-50 on ImageNet using unstructured sparsity scheme.

| Method | Training FLOPS ($\times$e18) | Inference FLOPS ($\times$e9) | Top-1 Accuracy (%) | Training FLOPS ($\times$e18) | Inference FLOPS ($\times$e9) | Top-1 Accuracy (%) |
|---|---|---|---|---|---|---|
| Dense | 4.8 | 8.2 | 76.9 | | | |
| Sparsity ratio | | 80% | | | 90% | |
| SNIP [9] | 1.67 | 2.8 | 69.7 | 0.91 | 1.9 | 62.0 |
| GraSP [10] | 1.67 | 2.8 | 72.1 | 0.91 | 1.9 | 68.1 |
| DeepR [15] | n/a | n/a | 71.7 | n/a | n/a | 70.2 |
| SNFS [12] | n/a | n/a | 73.8 | n/a | n/a | 72.3 |
| DSR [13] | 1.28 | 3.3 | 73.3 | 0.96 | 2.5 | 71.6 |
| SET [14] | 0.74 | 1.7 | 72.6 | 0.32 | 0.9 | 70.4 |
| RigL [11] | 0.74 | 1.7 | 74.6 | 0.39 | 0.9 | 72.0 |
| RigL$_{5\times}$ [11] | 3.65 | 1.7 | 76.6 | 1.95 | 0.9 | 75.7 |
| MEST$_{0.5\times}$+EM&S | **0.74** | 1.7 | **75.11** | **0.39** | 0.9 | **72.37** |
| MEST$_{0.67\times}$+EM | **0.74** | 1.7 | **75.39** | **0.39** | 0.9 | **72.58** |
| MEST$_{0.5\times}$+EM&S+DE | **0.70** | 1.7 | **75.09** | **0.37** | 0.9 | **72.36** |
| MEST+EM | **1.10** | 1.7 | **75.75** | **0.48** | 0.9 | **73.63** |
| MEST+EM&S | **1.27** | 1.7 | **75.73** | **0.65** | 0.9 | **75.00** |
| MEST+EM&S+DE | **1.17** | 1.7 | **75.70** | **0.60** | 0.9 | **75.10** |
| MEST$_{1.7\times}$+EM | **1.84** | 1.7 | **76.71** | **0.80** | 0.9 | **75.91** |
| MEST$_{1.7\times}$+EM&S | **2.15** | 1.7 | **77.19** | **1.11** | 0.9 | **76.13** |
| MEST$_{1.7\times}$+EM&S+DE | **1.96** | 1.7 | **77.11** | **1.01** | 0.9 | **76.08** |

training FLOPS can be further reduced while maintaining the same accuracy as using the full dataset. Note that the RigL exploits the dense model gradients to dynamically select the model structure during the sparse training process, which requires frequent dense backpropagations to calculate the dense gradients, and it is not memory-economic for Edge devices. The more analysis and results for ResNet-34 are shown in Appendix H.

**Exploring Sparsity Schemes.** Figure 5 (a) and (b) illustrates the accuracy by different sparsity schemes, i.e., unstructured, structured (channel), and fine-grained structured (block and pattern) when using our MEST+EM and MEST+EM&S, respectively. In block-based sparsity scheme, we set the block size as $(4, 1)$. And in pattern-based sparsity scheme, we use 8 sparse patterns according to [36].

We evaluate MEST framework with a sparsity ratio ranging from 10% to 98%. Note that due to structural nature of pattern-based pruning, its sparsity ratio must be at least 55.6% (see Appendix A). To make a fair comparison, we only choose the reference works that can maintain both sparse weights and gradients along the entire training process. Figure 5 (a) and (b) shows that all other sparsity schemes outperform channel sparsity scheme as expected, but the accuracy of channel sparsity scheme can be improved with weight elastic mutation. Our MEST+EM results show that the accuracy of our unstructured scheme results are higher than reference works and our block-based and pattern-based scheme results under all sparsity. The block-based and pattern-based schemes achieve similar accuracy as our unstructured scheme when sparsity is lower than 80%. With our MEST+EM&S, the accuracy of all sparsity schemes are boosted and outperforms the reference works.

## 5.2 Memory Footprint and Training Acceleration by Sparsity and Data-Efficient Training

In Figure 5 (c), we compare the model accuracy, training acceleration, and memory footprint among our MEST and representative SOTA works, i.e., SET, DSR, SNIP, and GraSP. We show the results using ResNet-32 on CIFAR-100 with 90% sparsity. The training acceleration results are normalized to dense training. The area of the circles represents the relative costs of the memory footprint (the smaller the better).

With unstructured sparsity, all reference works and our unstructured sparsity (without DE) results can only achieve minor training acceleration ($1.04\times \sim 1.27\times$), even under such a high sparsity ratio. Because the unstructured sparsity leads to irregular memory access which introduces significant execution overhead. Moreover, the dense model can take advantage of Winograd [48] to significantly accelerate the computation speed, but cannot be applied to sparse model. For our block-based and pattern-based sparsity schemes, without applying our data-efficient training, the acceleration

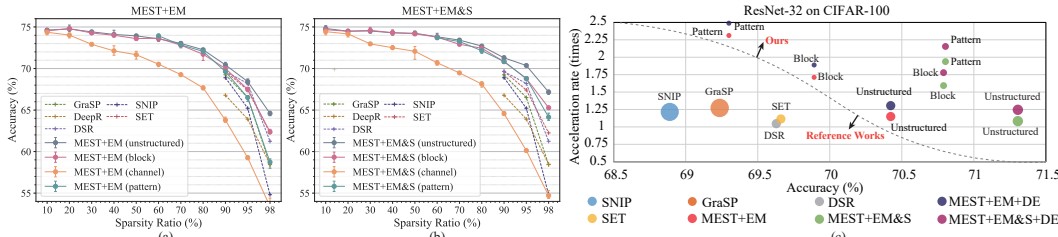

Figure 5: Results obtained using ResNet-32 on CIFAR-100. We choose the reference works that can maintain both sparse weights and gradients along the entire training process. (a) and (b) Accuracy results of the proposed MEST framework using different sparsity schemes and sparsity. (c) Comparison with representative SOTA works in accuracy, training acceleration rate, and memory footprint. Sparsity ratio is 90% for all results. The acceleration rate is normalized with respect to dense training. The size of the circles represents the relative cost of the memory footprint. We use $th = 0$ as the threshold for DE.

rates are greatly increased and achieve up to $2.3\times$ acceleration compared to the dense training while maintaining comparable accuracy. Our data-efficient training approach can effectively provide an extra speedup to all our results. The speedup is from 10% to 15% while not compromising the accuracy. The acceleration from data-efficient training is much higher on the CIFAR-10 dataset ($20.6\%\sim22.5\%$) since more examples are unforgettable and can be removed (more details in Appendix H). Comparing with SNIP, GraSP, SET, and DSR, the best-performant MEST increases accuracy by 1.91%, 1.54%, 1.14%, and 1.17%; achieves $1.76\times$, $1.65\times$, $1.87\times$, and $1.98\times$ training acceleration rate; and reduces the memory footprint by $8.4\times$, $8.4\times$, $1.2\times$, and $1.2\times$, respectively.

From the memory footprint perspective, SNIP and GraSP involve dense model during pruning at initialization, their memory footprints are considered the same as the dense model. Since the fewer indices are needed, the block-based sparsity and pattern-based sparsity under both MEST+EM and MEST+EM&S methods achieve a smaller memory footprint than all reference works and our unstructured sparsity scheme. More detailed discussion is in Appendix B. And a discussion about why it is critical to be memory-economic in edge training is shown in Appendix J.

Our results show a clearer advantage compared to the reference works. Even without DE acceleration, when using MEST+EM with block-based or pattern-based sparsity, or using MEST+EM&S with block-based sparsity, our results still outperform all reference works in all accuracy, training acceleration, and memory footprint aspects.

**Discussion.** The pattern-based sparsity shows a consistently better performance in acceleration than block-based sparsity. However, the accuracy comparison between pattern-based sparsity and block-based sparsity is varied according to the sparsity ratio and dataset. Since the pattern-based sparsity is only applicable to $3\times3$ CONV layers, for the network with different types of layers, a layer-wise sparsity scheme assignment is desired. Investigations of hybrid sparsity schemes are provided in Appendix K.

On the other hand, since both dataset compression and model sparsity make the trade-off between accuracy and acceleration, it is interesting to investigate the performance of different combinations of these two methods (more details in Appendix L).

## 6  Conclusion

This paper proposes a Memory-Economic Sparse Training (MEST) framework with enhancements by Elastic Mutation and Soft Memory Bound Elastic Mutation. Then, this paper systematically investigates the sparse training problem with respect to the sparsity schemes. We implement a prototype design on a mobile device to accurately measure the training speed performance. We investigate and incorporate the data-efficient training in sparse training scenario to further boost the acceleration. With MEST framework, a feasible solution is provided for sparse training on edge devices with superior performance on accuracy, speed, and memory footprint.

## Acknowledgment

This work is partly supported by the National Science Foundation CCF-1937500, CCF-1733701, and CCF-2047516 (CAREER), Army Research Office/Army Research Laboratory via grant W911NF-20-1-0167 (YIP) to Northeastern University, and Jeffress Trust Awards in Interdisciplinary Research to William & Mary. Any opinions, findings, and conclusions or recommendations expressed in this material are those of the authors and do not necessarily reflect the views of NSF, ARO, or Thomas F. and Kate Miller Jeffress Memorial Trust.

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
