# Appendix

## A   Sparsity Scheme

We use Figure A.1 to illustrate existing weight sparsity schemes, where grey represents the zero weights and colored parts are for remaining non-zero weights in a sparse model. In Figure A.1 (a)~(c), the GEMM matrix format is used for demonstrating the sparsity schemes. Figure A.1 (d) illustrates directly on the weight tensor.

Figure A.1 (a) is the **unstructured sparsity** [16, 1], where zero weights are distributed at arbitrary locations. The unstructured sparsity can achieve a high sparsity ratio with negligible effect on accuracy, but is not compatible with data-parallel executions on computing devices.

Figure A.1 (b) shows a type of **structured sparsity** [24, 2, 25, 26, 17, 3, 27, 28, 29, 30, 31, 18], called channel sparsity, where the weights of entire channels are set to zeros. The other type of structured sparsity is the filter sparsity [24, 28, 30, 29]. These two types of structured sparsity are somewhat equivalent, because if some filters are removed in one layer, the corresponding channels of the next layer become invalid. The structured sparsity preserves regularity on the sparse models, but suffers from significant accuracy loss.

Figure A.1 (c) and (d) show two types of fine-grained structured sparsity [19, 34, 35, 36, 37]: block-based sparsity and pattern-based sparsity.

For **block-based sparsity**, weights are partitioned into blocks with the same size. Figure A.1 (c) illustrates an example block size of $(4, 1)$. In the block-based sparsity, weights in the same block are either set to zeros or remaining together. Since block-based sparsity uses a much finer granularity compared with structured sparsity, it is considered as a fine-grained structured sparsity.

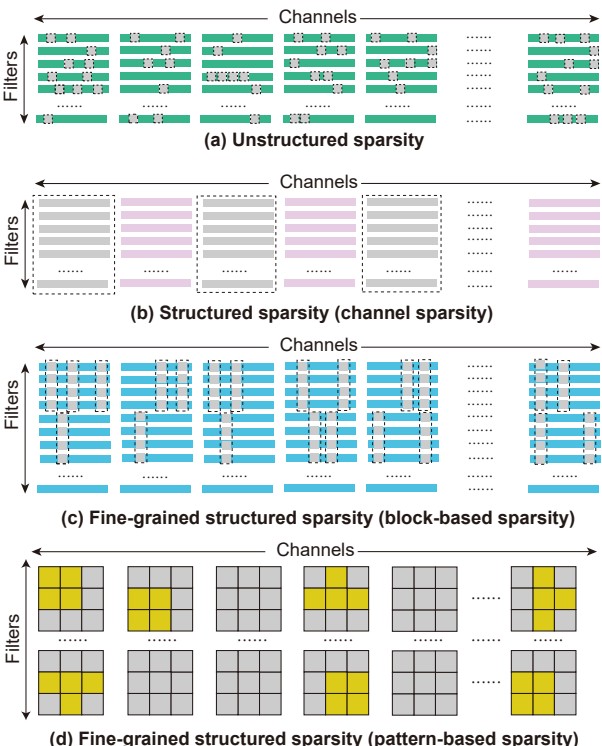

Figure A.1: (a) Unstructured sparsity; (b) structured sparsity (channel); (c) fine-grained structured sparsity (block); and (d) fine-grained structured sparsity (pattern).

Figure A.1 (d) shows the **pattern-based sparsity**, which is a combination of *kernel pattern sparsity* and *connectivity sparsity*. In kernel pattern sparsity, for each kernel in a filter, a fixed number of

weights are set to zeros, and the remaining weights form specific kernel patterns. The example in Figure A.1 (d) is defined as 4-entry kernel pattern, since every kernel preserves 4 non-zero weights out of the original $3\times3$ kernels. Besides that, the connectivity sparsity cuts the connections between some input and output channels, which is equivalent to removing corresponding whole kernels.

Table A.1: The impacts of sparsity schemes on the memory footprint in sparse training.

| Scheme | Memory Footprint | Approximation |
|---|---|---|
| Dense | $2N \cdot b_w$ | $2N \cdot b_w$ |
| Structured | $(1-s) \cdot (2N \cdot b_w)$ | $(1-s) \cdot (2N \cdot b_w)$ |
| Unstructured | $(1-s) \cdot (2N \cdot b_w + N \cdot b_{index}) + \sum_l ((F_l+1) \cdot b_{index})$ | $(1-s) \cdot (2N \cdot b_w + N \cdot b_{index})$ |
| Pattern | $(1-s) \cdot (2N \cdot b_w + \frac{1}{4}N \cdot b_{index}) + \sum_l ((F_l+1) \cdot b_{index})$ | $(1-s) \cdot (2N \cdot b_w + \frac{1}{4}N \cdot b_{index})$ |
| Block | $(1-s) \cdot (2N \cdot b_w + \frac{1}{B}N \cdot b_{index}) + \sum_l ((\frac{1}{m}F_l+1) \cdot b_{index})$ | $(1-s) \cdot (2N \cdot b_w + \frac{1}{B}N \cdot b_{index})$ |

# B  Memory Footprint

Consider a sparse model with a sparsity ratio $s \in [0, 1]$ obtained from a dense model with a total of $N$ weights. A higher value of $s$ denotes fewer non-zero weights in the sparse model. Suppose weights are represented as $b_w$-bit numbers. Each gradient is therefore represented with $b_w$ bits. For sparse models, we need indices for denoting the sparse topology of weights/gradients within the dense model. Indices are represented as $b_{index}$-bit numbers. Generally, mobile edge devices can support 8-bit fixed-point, 16-bit floating-point, and 32-bit floating-point numbers. Weights and gradients are usually using 16-bit or 32-bit. Due to the data storage format on edge devices, 8-bit or 16-bit is preferred for indices.

Table A.1 lists the memory footprint of sparse training in relevance to the sparsity scheme. In the **structured sparsity** scheme, where entire filters/channels are zeros, the sparse model can be reconstructed into a smaller dense model without indices. Thus, the memory footprint is determined by the number of non-zero weights plus the number of corresponding gradients i.e., $(1-s) \cdot 2N$.

For **unstructured sparsity** scheme, each non-zero weight and its gradient require indices to denote the corresponding location within the dense matrix. Compressed sparse row (CSR) format is commonly used for sparse storage. More specifically, consider the weights of a $l$-th CONV layer reshaped from 4-D tensor to a 2-D weight matrix, where each row represents the weights from a filter. We use $F_l$, $Ch_l$, and $K_l$ to denote the number of filters (output channels), number of channels (input channels), and kernel size, respectively. Thus, there are $F_l$ rows and $Ch_l \cdot K_l^2$ columns in a weight matrix. In CSR format, each non-zero weight requires a column index i.e., $col\_index$ to denote its location within the column. The number of column indices is equal to the number of non-zero weights. Also, the number of non-zero weights in each row (filter) should be denoted by $row\_index$, which is a vector with $F_l + 1$ elements. The difference between two adjacent elements in $row\_index$ denotes the number of non-zero weights in each row. Thus, the total number of indices of the entire network for the unstructured sparsity scheme is $(1-s) \cdot N + \sum_l (F_l+1)$. And the memory footprint of model representation together with gradients for unstructured sparsity is shown in Table A.1. Note that the number of filters (i.e., $F_l$) is much smaller than the total number of weights, and therefore the last term can be ignored as an approximate.

For **pattern-based sparsity** scheme, each kernel with non-zero weights requires a $kernel\_index$ to represent the kernel location in a filter. For the case of 4-entry kernel pattern used in Figure A.1 (d), it is equivalent to every 4 non-zero weights sharing a $kernel\_index$. Similar as the unstructured sparsity, the $row\_index$ is also needed.

For **block-based sparsity** scheme, consider the block size of $B = m \times n$, since all the weights within a block will be either zero or non-zero together, it is equivalent to every $B$ non-zero weights sharing a $block\_index$. As for the $row\_index$ vector, a total of $\frac{1}{m}F_l + 1$ elements are needed.

For the pruning-at-initialization algorithms, SNIP and GraSP involve several dense training iterations to determine the importance of initial weights. When under the end-to-end edge training scenario, their peak memory footprint is considered the same as the dense training: $2N \cdot b_w$.

On the other hand, the sparse training work such as SNFS [12] and RigL [11], which requires the sparse weights (unstructured) and dense gradients during the sparse training to evaluate the weight importance, the memory footprint is in between dense training and full sparse training and can be approximated as: $(2 - s) \cdot N \cdot b_w + (1 - s) \cdot N \cdot b_{index}$.

## C  Compiler-Level Optimizations for Training Acceleration

### C.1  How Does Sparsity Accelerate Training?

The training process consists of two phases, the *forward propagation* and the *backward propagation*. Considering the $l$-th convolutional (CONV) layer in the neural network, the forward propagation phase during training, which is the same as the inference process, can be formulated as:

$$a^l = \sigma\left(z^l\right) = \sigma\left(W^l * a^{l-1} + b^l\right),  \tag{3}$$

where $W$, $b$, and $z$ represent the weights, biases, and output before activation, respectively; $\sigma(\cdot)$ denotes the activation function; $a$ is the activations; $*$ means convolution operation. Many previous works [28, 30, 29, 34, 35, 37] have proved that the sparse weight matrices (tensors) can result in inference acceleration by effectively reducing the number of multiplications in convolution operation. Thus, the sparsity can inherently accelerate the forward propagation phase in the training process.

On the other hand, the goal of the backward propagation phase is to obtain the gradients of the weights, so as to update the weights. The two main calculation steps are as follows:

$$\delta^l = \delta^{l+1} * \text{rotate } 180^\circ \left(W^{l+1}\right) \odot \sigma'\left(z^l\right),  \tag{4}$$

$$G^l = a^{l-1} * \delta^l,  \tag{5}$$

where $\delta^l$ is the error associated with the $l$-th layer; and $G^l$ denotes the gradients. In the above equations, $\odot$ represents element-wise product, $\sigma'(\cdot)$ denotes the derivative of activation, and $\text{rotate } 180^\circ(\cdot)$ means rotating matrix 180 degrees.

It can be observed that the computations in both two steps are essentially based on convolution (i.e., matrix multiplication). The former uses sparse weight matrix (tensor) as the operand, and therefore can be accelerated in the same way as the forward propagation. The latter allows a sparse output result since the gradients have the same sparsity topology as the weights. Thus, both two steps have reduced computations, which are roughly proportional to the sparsity ratio, and therefore can be accelerated in the back propagation.

### C.2  Compiler Optimizations

In previous works such as PatDNN [35], compiler-level optimizations are used for accelerating inference. We extend those compiler optimizations and incorporate various sparsity schemes for accelerating the sparse training computation in both the forward-propagation and backward-propagation on edge devices. We adopt several compiler optimization techniques, including *sparse model storage*, *matrix reorder*, and *parameter auto-tuning* to relieve the poor memory performance, heavy control-flow instructions, thread divergence, and load imbalance caused by sparse computation, and thus achieving the sparse training acceleration.

**Sparse Model Storage.** Based on the CSR format for unstructured sparse model representation, we use more compact model storage formats delicately designed for pattern-based sparsity and block-based sparsity, which can better compress the storage for indices by leverage the structural regularity of the sparsity schemes and save memory-bandwidth of edge devices. Moreover, the data locality is further improved, enabling later branch-less execution.

**Matrix Reorder.** For the computation during the forward- and backward- propagation for each layer, the matrix multiplication is executed by multiple GPU threads simultaneously. Since the weight matrix is highly sparsified, and the non-zero weights are not evenly distributed across the whole weight matrix, the threads may execute the patterns/blocks with significantly divergent computations if the computation follows the original matrix order. Thus, we introduce the matrix reorder optimization to group the rows (filters) in the weight matrix that have similar computation patterns together (i.e., grouping the rows containing a similar number of non-zero patterns/blocks to be computed.) After

reordering the matrix, the rows in each group are assigned to multiple threads to achieve balanced processing.

**Parameter Auto-tuning.** Sparse training on edge devices involves many execution-related and performance-critical tunable parameters such as the memory data placement, matrix tiling sizes, looping unrolling factors, etc. These parameters will significantly vary the computation efficiency as well as the training speed. The best-suited configuration of the parameters is hard to be determined manually. Thus, We introduce the parameter auto-tuning technique to search the parameters in an automatic manner.

## D   Elastic Mutation for Pattern-based and Block-based Sparsity Schemes

Unlike the unstructured sparsity scheme, the mutation processes of pattern-based and block-based sparsity schemes are required to satisfy the structural constraints of those sparsity schemes.

**Pattern-based sparsity:** We perform `ArgRemoveTo(·)` by removing the least important convolution kernels to meet the sparsity ratio setting. Note that the importance of a specific convolution kernel can be obtained by summing up weight importance in the kernel. For `ArgGrowTo(·)`, we randomly select empty kernels (i.e., all weights in kernel are zeros) and set a random pattern style to them. The newly activated weights can be trained from their initial values, which are zeros. The total activated weights should meet the `ArgGrowTo(·)` sparsity setting. Please note that `ArgRemoveTo(·)` and `ArgGrowTo(·)` select the same number of kernels to remove or grow in a convolution filter within each layer for a balanced computation regime [34, 36].

**Block-based sparsity:** We perform `ArgRemoveTo(·)` by removing the least important blocks to meet the sparsity ratio setting. Note that the importance of a specific block can be obtained by summing up weigh importance in the block. For `ArgGrowTo(·)`, we randomly select empty blocks (i.e., all weights in block are zeros) and grow the whole block from zero values. The total activated weights should meet the `ArgGrowTo(·)` sparsity setting.

## E   Weight Importance Estimation

In MEST+EM(&S), besides the weight magnitude, we also consider the weight's current gradient as an indicator for its changing trend to estimate its importance. Because the weight with relatively large magnitude may become smaller, indicating it is becoming unimportant, while the small-magnitude weights can become larger as well. According to the Equation (1) in the main paper, we consider three types of weights are relatively important in our mutation process, which are the weights with 1) large weight magnitude but small gradient, 2) small weight magnitude but large gradient, and 3) large weight magnitude and large gradient.

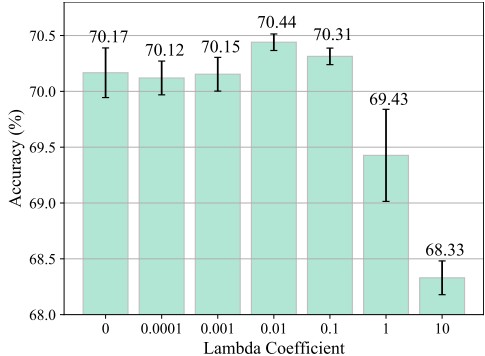

Figure E.1: MEST+EM accuracy with varying $\lambda$ coefficient using ResNet-32 on CIFAR-100.

Figure E.1 shows the sparse trained model accuracy under different $\lambda$ values. We use ResNet-32 on CIFAR-100 as an example. When $\lambda$ is set to 0.01 or 0.1, the accuracy can be improved compared to only considering the weight magnitude (i.e., $\lambda = 0$).

# F  Data-Efficient Training

## F.1  Basic Concepts

To explore the data efficiency in DNN training, measuring the effective information of a training sample for a network is a very important aspect. In [23], the number of forgetting events of a training example during the training process is used as an indicator to reflect the amount of information and the complexity of an example.

**Learning event.** A learning event occurs when a training sample goes from being misclassified to being correctly classified by a network in two consecutive training epochs.

**Forgetting event.** A Forgetting event occurs when a training sample goes from being correctly classified to being misclassified by a network in two consecutive training epochs.

**Unforgettable example.** Throughout the entire training process, if an example will never be misclassified after it has been correctly classified, the example is considered an unforgettable example. The examples that have never been correctly classified are not considered to be unforgettable.

According to prior works [20, 21, 22, 23], the unforgettable examples are generally considered as less informative and easy to be learned. The figure shows a example of unforgettable training examples and training examples with the highest forgetting event counts obtained using ResNet-32 on CIFAR-10 dataset. It can be observed that the unforgettable examples are intuitively much easier to recognize, which preserve distinctive object features and the objects have high contrast to the background. On the contrary, the most forgettable examples are clearly more complex compared to the unforgettable examples.

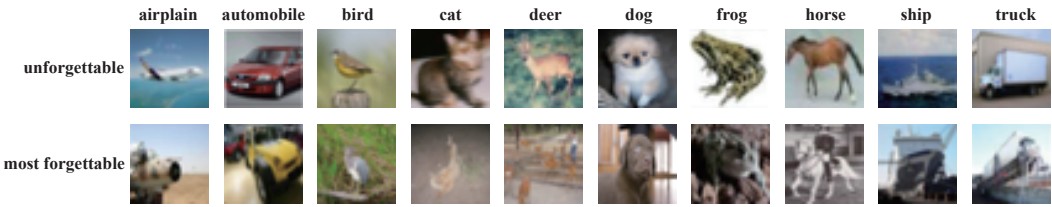

Figure F.1: Visualization of unforgettable examples and the most forgettable examples obtained using ResNet-32 on CIFAR-10.

## F.2  More Results for Final Accuracy using Different Number of Phase-1 Epochs

Figure F.2 shows the final accuracy status obtained using different number of first training phase epochs. The results include ResNet-32 on CIFAR-100, VGG-19 on CIFAR-10, and VGG-19 on CIFAR-100. The yellow grids stand for using that number of epoch for the first training phase can achieve similar accuracy as using a full dataset for the entire training process. Compared to the results on the CIFAR-10 dataset, the networks generally require more first training phase epochs on the CIFAR-100 to achieve similar accuracy as the full dataset training. This phenomenon can be observed for all pretraining and sparse training methods.

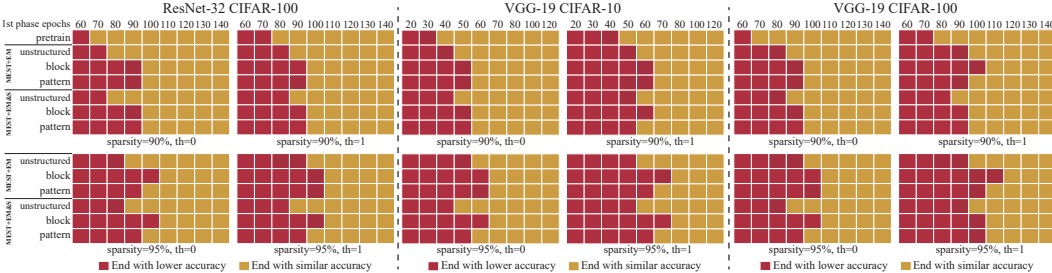

Figure F.2: The epoch number used for the first training phase and its corresponding final accuracy status under sparsity ratios of 90% and 95% and threshold of 0 and 1.

### F.3 Impact of Mutation on Unforgettable Examples

Figure F.3 shows the trend of the number of unforgettable examples throughout the entire training process, and we name it the forgetting curve. Note that different from the MEST(vanilla) method, the vanilla method here stands for a static sparse training method, which randomly pruned weights at initialization and without any mutation along the training process. Compared to the methods without mutation (i.e., pretrain and vanilla), the forgetting curve of the mutation methods (i.e., MEST+EM and MEST+EM&S) do not show severe fluctuations throughout the entire training process, indicating that our mutation method will not cause a notable increase in forgetting. This is because the mutated weights are least important, which only have a minor impact on the model performance. And our elastic mutation also gradually decreases the mutation rate, which further enhance the network stability. We can also observe that our MEST methods can increase the number of unforgettable examples compared to the vanilla method, which provides the potential of removing a larger portion of training examples and hence a higher acceleration.

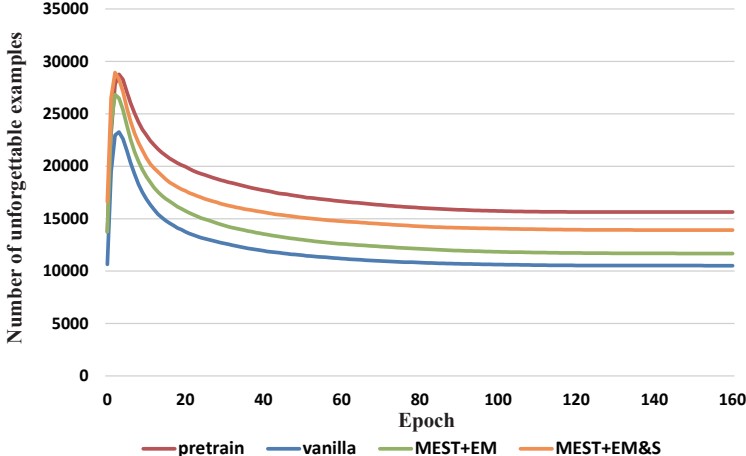

Figure F.3: The trend of the number of unforgettable examples throughout the entire training process.

# G   Experiment Setup

We list hyperparameter settings for the proposed MEST+EM and MEST+EM&S in Table G.1.

Table G.1: Hyperparameter settings.

| Experiments | VGG-19 on CIFAR | | ResNet-32 on CIFAR | | ResNet-50/34 on ImageNet | |
|---|---|---|---|---|---|---|
| Regular hyperparameter settings | | | | | | |
| Training epochs ($\tau_{end}$) | 160 | | 160 | | 150 | |
| Batch size | 64 | | 64 | | 2048 | |
| Learning rate scheduler | cosine | | cosine | | cosine | |
| Initial learning rate | 0.1 | | 0.1 | | 2.048 | |
| Ending learning rate | 4e-8 | | 4e-8 | | 0 | |
| Momentum | 0.9 | | 0.9 | | 0.875 | |
| $\ell_2$ regularization | 5e-4 | | 1e-4 | | 3.05e-5 | |
| Warmup epochs | 5 | | 0 | | 8 | |
| MEST hyperparameter settings | | | | | | |
| Number of epochs do EM ($\tau_{stop}$) | 130 | | 130 | | 120 | |
| EM frequency ($\Delta\tau$) | 5 | | 5 | | 2 | |
| MEST+EM schedule `ArgRemoveTo` sparsity (RM) `ArgGrowTo` sparsity (GR) | 0 - 100: 100 - 130: 130 - 160: | RM (s + 0.05) GR (s) RM (s + 0.025) GR (s) No EM | 0 - 100: 100 - 130: 130 - 160: | RM (s + 0.05) GR (s) RM (s + 0.025) GR (s) No EM | 0 - 90: 90 - 120: 120 - 150: | RM (s + 0.05) GR (s) RM (s + 0.025) GR (s) No EM |
| MEST+EM&S schedule `ArgRemoveTo` sparsity (RM) `ArgGrowTo` sparsity (GR) | 0 - 100: 100 - 130: 130 - 160: | GR (s - 0.05) RM (s) GR (s - 0.025) RM (s) No EM | 0 - 100: 100 - 130: 130 - 160: | GR (s - 0.05) RM (s) GR (s - 0.025) RM (s) No EM | 0 - 90: 90 - 120: 120 - 150: | GR (s - 0.05) RM (s) GR (s - 0.025) RM (s) No EM |
| Importance coefficient ($\lambda$) | 0.01 | | 0.01 | | 0.001 | |

# H   Accuracy Results of ResNet-32 on CIFAR-10, VGG-19 on CIFAR-10 and CIFAR-100, and ResNet-34 on ImageNet-2012

The MEST accuracy results for ResNet-32 on CIFAR-10 is shown in Figure H.1. When incorporating the data-efficient (DE) training, $29.5\% \sim 30\%$ training examples are removed without decreasing the accuracy. We also validate our MEST on VGG-19 using CIFAR-10 and CIFAR-100. The results are shown in Table H.1 and Figure H.2. For the ImageNet dataset, we also show MEST accuracy results on ResNet-34. Since the size of the ImageNet dataset itself is about 150GB and a ResNet50 requires more than 1 day to train on a 8 x 2080Ti GPU server, it is impractical to be trained on current edge devices such as mobile phones. Therefore, we mainly use ImageNet dataset to show the accuracy results and validate the effectiveness of our MEST. Table H.2 shows the accuracy results with both regular training effort (i.e., 150 epochs) and $1.7\times$ effort (i.e., 250 epochs). When incorporating the data-efficient (DE) training, $9.3\% \sim 10.7\%$ training examples are removed.

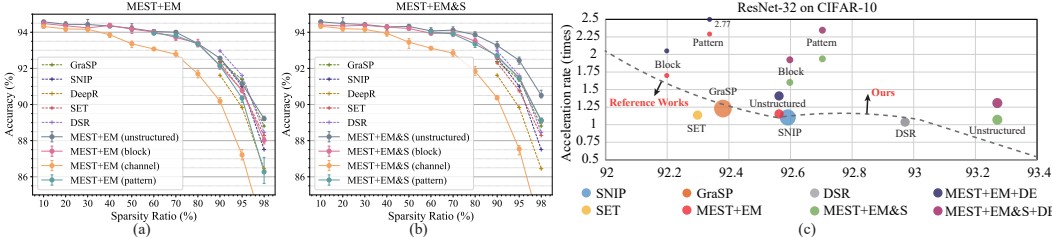

Figure H.1: Results obtained using ResNet-32 on CIFAR-10. We choose the reference works that can maintain both sparse weights and gradients along the entire training process. (a) and (b) Accuracy results of the proposed MEST framework using different sparsity schemes and sparsity. (c) Comparison with representative SOTA works in accuracy, training acceleration rate, and memory footprint. Sparsity ratio is 90% for all results. The acceleration rate is normalized with respect to dense training. The size of the circles represents the relative cost of the memory footprint. We use $th = 0$ as the threshold for DE.

Table H.1: Accuracy comparison with SOTA works using VGG-19 on CIFAR-10 and CIFAR-100.

| Dataset | Memory Footprint | CIFAR-10 (*Dense: 94.20*) | | | CIFAR-100 (*Dense: 74.17*) | | |
|---|---|---|---|---|---|---|---|
| Sparsity ratio | | 90% | 95% | 98% | 90% | 95% | 98% |
| LT [49] | dense | 93.51 | 92.92 | 92.34 | 72.78 | 71.44 | 68.95 |
| SNIP [9] | dense | 93.63 | 93.43 | 92.05 | 72.84 | 71.83 | 58.46 |
| GraSP [10] | dense | 93.30 | 93.04 | 92.19 | 71.95 | 71.23 | 68.90 |
| DeepR [15] | sparse | 90.81 | 89.59 | 86.77 | 66.83 | 63.46 | 59.58 |
| SET [14] | sparse | 92.46 | 91.73 | 89.18 | 72.36 | 69.81 | 65.94 |
| DSR [13] | sparse | 93.75 | 93.86 | 93.13 | 72.31 | 71.98 | 70.70 |
| MEST (vanilla) | sparse | 91.73±0.27 | 90.43±0.33 | 87.82±0.27 | 68.57±0.38 | 65.01±0.32 | 60.88±0.36 |
| MEST+EM | sparse | 93.07±0.49 | 92.59±0.36 | 90.55±0.37 | 71.23±0.33 | 69.08±0.46 | 64.92±0.42 |
| MEST+EM&S | sparse | 93.61±0.36 | 93.46±0.41 | 92.30±0.44 | 72.52±0.37 | 71.21±0.41 | 69.02±0.34 |

Table H.2: Accuracy with ResNet-34 on ImageNet.

| Method | Top-1 accuracy (%) | | | |
|---|---|---|---|---|
| Dense | 74.08 | | | |
| Sparsity ratio | 60% | 70% | 80% | 90% |
| MEST (vanilla) | 73.08 | 70.71 | 69.74 | 65.68 |
| MEST+EM | 74.10 | 73.66 | 72.83 | 70.38 |
| MEST+EM&S | 74.12 | 73.81 | 73.57 | 72.10 |
| MEST+EM&S+DE | 74.17 | 73.83 | 73.48 | 72.03 |
| MEST$_{1.7\times}$ (vanilla) | 73.21 | 70.79 | 69.92 | 65.77 |
| MEST$_{1.7\times}$+EM | 74.30 | 73.89 | 73.12 | 70.76 |
| MEST$_{1.7\times}$+EM&S | 74.34 | 73.97 | 73.86 | 72.25 |
| MEST$_{1.7\times}$+EM&S+DE | 74.37 | 73.93 | 73.79 | 72.13 |

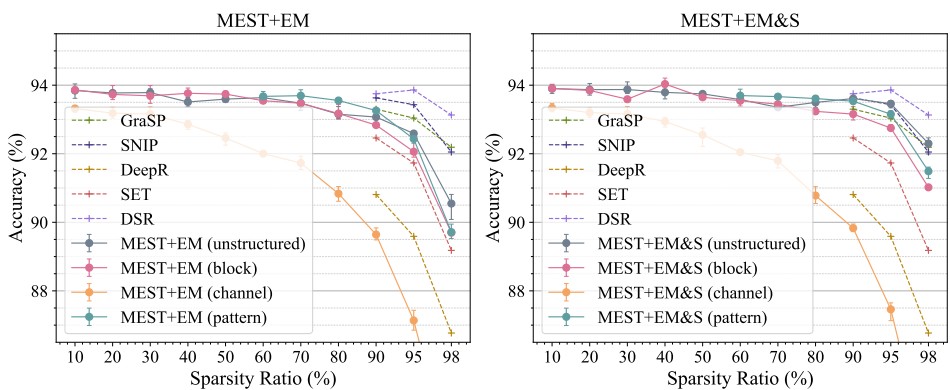

(a) Accuracy comparison on VGG-19 using CIFAR-10 dataset.

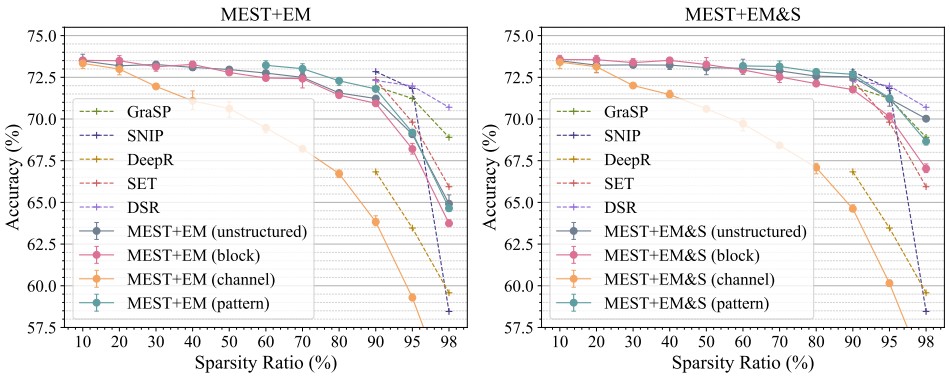

(b) Accuracy comparison on VGG-19 using CIFAR-100 dataset.

Figure H.2: Accuracy of the proposed MEST framework using different sparsity schemes on VGG-19.

# I MEST Accuracy on Compact Model MobileNet-V2 and Deeper Model ResNet-110

We also evaluate our MEST+EM&S on MobileNet-V2 as the representative of compact models and on ResNet-110 as the representative of deep models. Table I.1 shows the accuracy achieved by our MEST+EM&S method on CIFAR10 under different sparsity ratios, and Table I.2 shows the overall training FLOPs ($\times$e12) and number of model parameters (M). The MobileNet-V2 is more sensitive to sparsity compared to ResNet-32 and ResNet-110, as shown in Table I.1. Actually, the ResNet-32 (as well as ResNet-20, ResNet-110) are the lightweight ResNet version dedicated to CIFAR tasks, while the ResNet-18, ResNet-34, and ResNet-101 are the large versions for the ImageNet. So, as we can see from Table I.2, the ResNet-32 is even smaller than MobileNet-V2 (1.86M parameters v.s. 2.3M parameters.), while the computation cost of ResNet-32 is higher than MobileNet-V2 due to the depth-wise separable CONV. Note that the ResNet-32 here (and in the paper) is a ($2\times$) widened version, which is consistent with the reference works cited in the paper (all as shown in Table 1). This is the reason that the number of parameters and training cost of ResNet-32 is similar to the ResNet-110 as shown in Table I.2.

It is interesting to see that, under 90% sparsity, ResNet-32 has a similar accuracy and training FLOPs as the MobileNet-V2 under 60% sparsity, while the number of parameters of ResNet-32 is $4.8\times$ less than MobileNet-V2. For this case, the ResNet-32 will be more desired than MobileNet-V2. Moreover, MobileNet-V2 is much deeper (57 CONV layers) than ResNet-32, which will require more data movement among memory and cache for reading and writing intermediate results and lead to a higher execution overhead.

Table I.1: Accuracy comparison on ResNet-32, MobileNet-V2, and ResNet-110 on CIFAR-10 using MEST+EM&S.

| Sparsity | Dense | 50% | 60% | 70% | 80% | 90% |
|---|---|---|---|---|---|---|
| ResNet-32 | 94.88 | 94.41 | 94.05 | 94.14 | 93.70 | **93.27** |
| MobileNet-V2 | 94.08 | 94.06 | **93.32** | 93.05 | 92.38 | 90.61 |
| ResNet-110 | 94.64 | 93.47 | 93.73 | 93.62 | 93.26 | **92.29** |

Table I.2: Comparison of train FLOPs ($\times$e12) and number of parameters (M) on ResNet-32, MobileNet-V2, and ResNet-110 on CIFAR-10 using MEST+EM&S.

| Sparsity | Dense | 50% | 60% | 70% | 80% | 90% |
|---|---|---|---|---|---|---|
| ResNet-32 | 6.38 / 1.86 | 3.30 / 0.93 | 2.68 / 0.74 | 2.07 / 0.56 | 1.45 / 0.37 | **0.83 / 0.19** |
| MobileNet-V2 | 2.11 / 2.30 | 1.09 / 1.15 | **0.88 / 0.92** | 0.68 / 0.69 | 0.48 / 0.46 | 0.28 / 0.23 |
| ResNet-110 | 5.74 / 1.70 | 2.97 / 0.85 | 2.41 / 0.68 | 1.86 / 0.51 | 1.31 / 0.34 | **0.75 / 0.17** |

# J Why Does Memory-Economic Critical for Training on Edge Devices?

The availability of edge devices for training requires consideration of two aspects: (1) whether the dataset and model can be accommodated by a mobile device; (2) whether the free space of device memory (RAM) is sufficient for the required training memory footprint.

The current mobile devices generally have memory in GB levels. For example, current general mobile devices such as Samsung Galaxy A20s, Google Pixel 3, and Samsung S20 have 2GB or more memory. However, unlike the training on a high-end GPU cluster where all the memory can be reserved for training, the memory on mobile devices will also be partially occupied by the operating system and other backend applications. This puts an even greater strain on the memory of mobile devices. Thus, having a smaller memory footprint will always benefit the training on mobile devices.

We use ResNet-32 and VGG-19 as examples. Assume 32 bits weights, 8 bits indices are used with batch size of 64. For ResNet-32, the dense model and the methods that involve dense computations require a 462MB memory footprint for model size, while our sparse model with unstructured sparsity requires 46MB (69MB) and 23MB (46MB) under 90% and 95% sparsity for MEST+EM(&S), respectively. For VGG-19, a significant reduction can be obtained, where the dense model requires 4964MB memory footprint, while our sparse model with unstructured sparsity requires 494MB (744MB) and 247MB (494MB) under 90% and 95% sparsity for MEST+EM(&S), respectively.

MEST successfully reduces the memory footprint (to less than 100M and 800M for ResNet-32 and VGG-19) while maintaining a similar or higher accuracy than prior works, while the reference methods either require a dense memory footprint (LT, SNIP, GraSP, RigL) or suffer a severer accuracy degradation.

# K  Layer-wise Sparsity Scheme and Sparsity Ratio

As shown in the main paper, different sparsity schemes have different performance in accuracy and acceleration rate. Moreover, pattern-based sparsity is only applicable to 3×3 CONV layers, while many popular networks contain a large portion of 1×1 CONV layers or FC layers. Therefore, hybrid sparsity schemes may be a better option, although the pattern-based sparsity is still preferred for 3×3 CONV layers. On the other hand, different types and different sizes of layers inherently exhibit different weight redundancy and therefore deserve non-uniform sparsity ratios among layers.

In this section, we further investigate the performance when using a layer-wise sparsity scheme and sparsity ratio assignment. We use ResNet-50 on CIFAR-100 as an example, and Table K.1 shows the results of accuracy and training speed when using a single sparsity scheme or using different sparsity schemes on different types of layers. The training speed is the time of a training iteration in seconds while using a batch size of 64 and measured on a Samsung smartphone using the mobile GPU. For ResNet-50, over 50% of weights and computations are contributed by the 1×1 CONV layers. If we only adopt pattern-based sparsity, which is only applicable to 3×3 CONV layers, the overall sparsity ratio is 44% under a 90% sparsity on the 3×3 CONV layers. Therefore, only use pattern-based sparsity is not able to achieve a high sparsity ratio and hence lower training acceleration. Block-based sparsity can be adopted across the entire network. But both the accuracy and training speed are lower than using hybrid sparsity schemes, i.e., adopting pattern-based sparsity to 3×3 CONV layers and block-based sparsity to 1×1 CONV layers, respectively.

We also explore different sparsity ratio strategies by comparing three sparsity ratio settings, including 1) the uniform sparsity ratio (90%) on all the layers, 2) a fixed ratio (1.12:1) between 3×3 CONV layers and 1×1 CONV layers in the entire network (i.e., 95% pattern-based sparsity for all 3×3 CONV layers and 85% block-based sparsity for 1×1 CONV layers), and 3) a layer-wise ratio assignment proportional to the layer size. All three strategies are hybrid sparsity schemes and have the same overall sparsity ratio (90%).

Table K.1: Comparison of accuracy and training speed using different sparsity schemes and different layer-wise sparsity ratios on ResNet-50 using CIFAR-100.

| Scheme | Sparsity Ratio | Accuracy (%) | Training Speed ($s/iter$) |
|---|---|---|---|
| Dense | 0% | 77.18 | 11.92 |
| MEST+EM | | | |
| Pattern | 44% (90%) | 75.36 | 8.18 |
| Block | 90% | 72.82 | 6.25 |
| Hybrid (uniform) | 90% | 72.87 | 5.39 |
| Hybrid (1.12:1) | 90% | 73.24 | 5.55 |
| Hybrid (proportional) | 90% | 73.56 | 5.62 |
| MEST+EM&S | | | |
| Pattern | 44% (90%) | 75.88 | 8.36 |
| Block | 90% | 73.68 | 6.79 |
| Hybrid (uniform) | 90% | 73.72 | 5.99 |
| Hybrid (1.12:1) | 90% | 73.98 | 6.08 |
| Hybrid (proportional) | 90% | 74.12 | 6.15 |

It can be observed that the three strategies have similar training speed, where the uniform sparsity ratio is the fastest since the acceleration rate is not linearly increased along with the increased sparsity ratio, as shown in Figure 2 in the main paper. On the other hand, when using non-uniform sparsity ratios, the accuracy can be improved, which indicates the larger layers have more redundant weights compared to smaller layers, and can tolerant a higher sparsity ratio.

## L Combinations of Dataset Compression and Model Sparsity

In our work, we intend to incorporate dataset-efficient training on top of the sparse training and without further decreasing the accuracy. However, when a minor accuracy drop is allowed, selecting the best-suited combination of dataset compression and model sparsity to achieve a higher acceleration while maintaining a higher accuracy is an interesting topic that can be further studied.

Table L.1: Comparison of accuracy results on different dataset compression and model sparsity combinations. The results are obtained by using MEST+EM&S with unstructured sparsity on ResNet-32 and CIFAR-10 dataset.

| Scheme | baseline | ① | ② |
|---|---|---|---|
| Sparsity | 90% | 95% | 90% |
| Removed examples | 0 | 0 | 17900 |
| Phase-1 epochs | - | - | 40 |
| Final accuracy (%) | 93.27 | 92.44 | 93.02 |

For example, as shown in Table L.1, we consider the MEST+EM&S result under 90% unstructured sparsity as the baseline scheme. If we intend to further increase the acceleration rate, we can choose two different schemes, including ① further increasing the sparsity or ② incorporating data-efficient training (i.e., compress the dataset). Based on our measurements, the scheme ① and ② provide the same acceleration rate (1.27×) on top of the baseline scheme. But we observe that incorporating data-efficient training instead of further increasing the model sparsity can deliver higher accuracy. Therefore, we may hypothesize that when the model sparsity goes beyond a certain degree, to further increase the acceleration rate while preserving a higher accuracy, it is more desired to compress the dataset than compress the model. Since both the number of removed examples and the removing epoch will affect the final model accuracy, it is a complicated problem that is worth to be further studied in the future.