# OpenReview forum: "MEST: Accurate and Fast Memory-Economic Sparse Training Framework on the Edge"
_NeurIPS.cc/2021/Conference — NeurIPS 2021 Spotlight_

### Official Review · Reviewer_KRc4 · 2021-07-16

**Rating:** 8
**Confidence:** 4

**Summary:**

This paper is conceived in the true spirit of sparse training (i.e., start with a sparse neural network; train at any moment in time  a sparse neural network and don’t use information from non-existing connections; do inference with the trained sparse neural network) to enable neural networks training at the edge on low-resource devices. In order to do so, the paper proposes a new sparse training algorithm with dynamic sparsity, named MEST, with a couple of variations for various settings (e.g., hard or soft memory constraints). Also, MEST, in contrast with the typical sparse training algorithms,  introduces the magnitude combined with the gradient as a metric to detect unimportant connections. Nevertheless, the paper is also considering data efficiency for the first time in sparse training (up to my knowledge). The extensive and rigorous empirical validation shows that MEST and its variants reach state-of-the-art performance in sparse training. Notably, even if it is not one of the paper claims, it seems that MEST can outperform dense training on convolutional neural networks using just half of the computational resources. Convolutional layers being perhaps the last type of neural network layer on which it is still arguable in the literature if truly sparse training can outperform dense training (see for details Table 2, left column, ResNet-50 on ImageNet).

**Limitations And Societal Impact:**

While within the whole paper its limitations and advantages seem fairly discussed, a self-contained paragraph where the limitations are clearly enumerated is missing. In my opinion, adding such a paragraph will just increase the paper quality.

**Main Review:**

**Originality**

This paper has an excellent level of originality and proposes several innovations on several dimensions for various settings which improves state-of-the-art performance. Remarkable, the proposed innovations are simple and efficient. Related work is well discussed. As minor comments:

_ I suggest changing the order of introducing Sparse Evolutionary Training (SET) and DeepR (e.g., lines 122-129), as SET has been released in 2017 with a couple of months before DeepR (see https://arxiv.org/abs/1707.04780).

_ The related work on fixed sparsity (Section 2.2.1) is missing few papers which have been published well before SNIP - Ref [9].

_ The related work on dynamic sparsity (Section 2.2.2) shall discuss also a method like Top-KAST (https://arxiv.org/abs/2106.03517) which is an improvement over RigL  - Ref [14] in terms of memory efficiency.

**Quality**

The submission and the proposed methods are technically sound. The empirical validation is well executed. For me, this looks like a complete solid piece of work.

**Clarity**

The paper has a very good structure, and it is clear and well written.  I believe that the paper together with the Appendix and the released code can help any interested reader to easily reproduce the results. As a minor comment, perhaps you can consider bringing some parts of Appendix E (e.g., Equation 5) to the main paper as it looks to me as an interesting contribution which may have a good influence on the results.

Actually, my only question for the rebuttal is related with Appendix E. Have you tried to assess systematically the influence of Equation 5 on the learning behavior? My hypothesis is that it helps MEST to speed up its learning process in comparison with other sparse training methods which use just magnitude to identify unimportant weights. It seems that Table 2 supports somehow this idea (see MEST 0.5x) and makes MEST to learn even faster than RigL in the context where MEST uses just random growth and not gradient growth (Am I right here?). Do you have the learning curves (e.g. training epoch versus accuracy - or similar) during training. If the space permit, it would be great if you can put one example of learning curves in the rebuttal, and perhaps more examples in the Appendix of the final paper.

**Significance**

I believe that this paper has the potential of becoming a landmark in sparse training with dynamic sparsity.


**Time Spent Reviewing:**

3

---

> ### Author Response · Authors · 2021-08-10
> **Author responses to reviewer KRc4**
>
> **Thank you for your thoughtful review. We will definitely reflect them in the paper to improve our work.**
>
> **Thanks for your acknowledgment of our contributions. One important motivation of our work is that we think of several factors that do not draw enough attention yet but are critical in sparse training, such as sparsity type, whether it involves dense computation, and data efficiency. We hope our work can bring some inspiration to the sparse training community.**
>
> ___
> ### **Q1. Rearrange the order of SET and DeepR, the missing references, and move some parts of Appendix E to the main paper.**
> Thank you for your constructive suggestions. We will reorder the introduction of SET and DeepR, and add the two reference works in the revision. We will also shrink Section 1 and Section 4 a little bit to put Equation 5 and the corresponding rationale to Section 3.
>
> ___
> ### **Q2. Does MEST use random growth? Learning speed comparison.**
> About your questions, yes, you are right. MEST is based on random growth since we want to avoid involving any dense computation and avoid using any dense information during the entire training process.
> In this paper, we were focused on the impact of Equation 5 on the final accuracy. But you do raise an interesting question that deserves further investigation.
>
> Since adding figures is not supported in the rebuttal, we sampled a few points from the training curve. RigL and MEST0.67x+EM results are following the same configurations as used in Table 2.
> Based on the results in Table A, we think your hypothesis is correct. Our MEST method shows a much faster learning speed compared to RigL, especially at the early and middle stages of training.
> We will further investigate this and provide more results and analysis. This can also be an interesting topic for future works.
>
> **Table A. Top-1 accuracy comparison of RigL and MEST on ImageNet using ResNet50 with uniform 90% unstructured sparsity.**
>
> >| epoch        | 5     | 20    | 35    | 50    | 65    | 80    | 100   |
> |--------------|-------|-------|-------|-------|-------|-------|-------|
> | RigL         | 22.57 | 30.98 | 59.95 | 59.43 | 68.65 | 70.15 | 72.0  |
> | MEST0.67x+EM | 34.81 | 52.90 | 66.64 | 65.66 | 70.23 | 70.83 | 72.58 |

---

> > ### Comment · Reviewer_KRc4 · 2021-08-26
> > **Rebuttal Response**
> >
> > I thank the authors for clarifying the raised issues. Congratulations for the very nice paper. I am really looking forward to see the outcome of the future works.

---

> > > ### Author Response · Authors · 2021-08-29
> > > **Author responses**
> > >
> > > Thank you for your appreciation of our work! We will revise our paper according to your suggestions. And thank you again for your valuable time.

---

### Official Review · Reviewer_wx9R · 2021-07-16

**Rating:** 7
**Confidence:** 3

**Summary:**

Training Deep Neural Networks on limited resource (memory/power) edge devices is a challenging problem considering the memory and computational complexity of the training algorithm. This paper presents, MEST, a novel memory economic DNN training framework which exploits model sparsity (scheme and ratio) to speed up the training process without significant accuracy loss.

The Authors present two improvements on the vanilla MEST algorithm in the form of Elastic mutation and soft memory bound which further enables increase in training speed while improving the training accuracy. Sparsity scheme and ratio are critical in determining the accuracy and training speedup derived from the proposed solution, the Authors discuss this dependence in detail.

Lastly, the paper also exploits data efficiency by capitalizing on unforgettable examples during training.  The Authors propose a strategy for identifying a set of unforgettable examples in-situ on the edge device and skipping it during part of the training process to improve training speed up.

The Authors present a detailed set of experiments for a few DNN models on a few target benchmarks while measuring accuracy, training speedup and memory footprint of the proposed algorithms.


**Limitations And Societal Impact:**

A quantitative comparison on the memory footprint costs of the prior art and proposed training algorithms would be handy to understand the tradeoffs between different MEST variants.

The selection of training time speedup doesn’t give clarity on whether the edge training devices used can support end-to-end DNN training with MEST algorithm. If such end-to-end training cannot fit on the edge device then the Authors’ claim that MEST makes training on the edge feasible might not be completely accurate.


**Main Review:**

The paper is well written and presents related work along with the key concepts utilized in the paper. This juxtaposition of past work and the proposed ideas improves readability and helps distinguish the new ideas.

The Authors propose MEST a novel DNN training algorithm that claims to support end-to-end training on edge devices. MEST utilizes sparsity schemes and ratios along with a magnitude + gradient weighted, weight mutation to train DNNs with speedup over dense baseline and small accuracy losses. They further add Elastic Mutation and Soft Memory Bound updates to the algorithm. These novel updates enhance the training speedup achieved by the vanilla algorithm with improvements in accuracy.

This paper also presents the first analysis on the combination of Data efficiency techniques for training speedup through skipping of unforgettable examples with dynamic sparse model-based training. The combination of the two approaches enables a shrinkage in memory footprint on top of MEST based approaches while adding more training speedup. The experiments conducted show the superiority of the MEST approaches in terms of improved accuracy, training speed and memory footprint.

In the introduction Authors use the term accuracy performance, this is confusing, and it would be great if the Authors could define it. Is it the product of accuracy and performance (training speed) or just accuracy?

Table 1 shows a comparison of MEST family of training algorithms and past approaches for training ResNet-32 on CIFAR-10/100. I don’t understand the authors choice of going with a qualitative description of the memory footprint for the different approaches instead of a quantitative metric. Quantitative description could help understand the added memory footprint impact of Soft Memory Bound optimization. This increase in memory footprint can be visualized in Figure 4c). If the Authors could clearly state, the expected memory footprint deterioration for utilizing the Soft Memory Bound update to MEST. This could help the users of MEST make an informed choice about trading off higher accuracy for said memory footprint cost.

Section 5 defines the training speed measurement as the computation time for forward/backward pass on a batch of 64 images. What is the significance of selecting this batch of images? The choice of 64 images seems quite low compared to the dataset required to achieve the accuracies quoted in the experiments. This begs the question that whether the MEST framework, despite showing solid improvements against past work, enables end-to-end training of target DNNs (ResNet-32, VGG-19 etc.) for the target datasets (CIFAR-10, CIFAR-100 …) on edge devices?

On multiple instances in the paper the Authors utilize the term “not practical/compatible for edge training” (e.g., line 306, 359). It would be helpful if the Authors could state what criteria limits the practicality/compatibility of these approaches. Specifically, what is the threshold of memory footprint identified by the Authors for edge devices. This question also impacts the interpretation of Figure 4c), which highlights the superiority of MEST solutions over prior art. Are MEST solutions small enough to fit on actual edge devices? Also, the choice of 90% sparsity ratio for figure 4c) seems to present the MEST+EM solution more optimistically than the 95% sparsity ratio where a few past approaches show comparable accuracy. Showing the worst case comparison for MEST will shed better light on MEST benefits.

In Section 3.1 the Authors correctly state that FLOPs don’t account for execution overheads from irregular memory access for sparse data ops. However, the Authors still utilize FLOPs as the normalizing factor for Table 2 to compare the accuracy benefits of MEST solutions compared to prior art. Could the Authors justify why its suitable to use FLOPs in this context?

Figure 2 elucidates the significance of selecting the correct sparsity scheme on the training speed. While channel sparsity provides significant speed improvements later results highlighted in Figure 4 a) and b) show that the accuracy penalty of channel sparsity is also the highest. Authors mention on line 349 that weight elastic mutation can help alleviate this poor accuracy result for channel sparsity. Do Authors have any results for this claim? Does this optimization bring channel sparsity within Figure 4 c)’s range?

To improve the readability of the results on accuracy, training speedup and memory footprint it would be nice if Authors reworded “the best-performant MEST” to “the best-performant MEST for the metric”. Since it gives the impression that a single configuration of MEST provides the reported improvements, which doesn’t seem obvious from the presented figures.


**Time Spent Reviewing:**

5

---

> ### Author Response · Authors · 2021-08-10
> **Author responses to reviewer wx9R (Part 1/2)**
>
> ### **Part 1**
>
> ___
> **Thanks for your time spent in the review. And your valuable comments and questions can significantly improve our work. We address them in the following and are very happy to add the clarifications in the revised paper.**
>
> ___
>
> ### **Q1: The meaning of “accuracy performance”**
>
> Sorry for the confusion. Accuracy performance means just the model accuracy. We’ll modify it in the revision.
>
> ___
> ### **Q2: Why does Table 1 use a qualitative description of memory footprint instead of a quantitative metric?**
>
> Thanks for this good point. We will use the quantitative description of memory footprint in Table 1 in the revised version.
>
> The reasons that we used “dense” and “sparse” are: (1) Memory footprint depends on the sparsity ratio, while each row in Table 1 uses multiple sparsity ratios. (2) In Table 1, we would like to make the point that even though some reference works are based on sparse training, they still require a dense memory footprint which is inefficient for sparse training on the edge.
>
> Here is an example of the memory footprint for the ResNet32 and VGG19 models with 90% (95%) sparsity, when 32-bit weights and 8-bit indices are used with a batch size of 64.
> Since LT, SNIP, and GraSP involve dense training, their memory footprint is the same as the dense model.
> We also provide details and analysis about how to calculate the memory footprint in Appendix B.
>
> **Table A. Memory footprint comparison using ResNet32 with 90% (95%) sparsity.**
>
> >|                                             | 90% sparsity | 95% sparsity |
> |---------------------------------------------|--------------|--------------|
> | LT, SNIP, and GraSP                         | 462MB        | 462MB        |
> | DeepR, SET, DSR, MEST(vanilla), and MEST+EM | 46MB         | 23MB         |
> | MEST+EM&S                                   | 69MB         | 46MB         |
>
>
> **Table B. Memory footprint comparison using VGG19 with 90% (95%) sparsity.**
>
> >|                                             | 90% sparsity | 95% sparsity |
> |---------------------------------------------|--------------|--------------|
> | LT, SNIP, and GraSP                         | 4964MB       | 4964MB       |
> | DeepR, SET, DSR, MEST(vanilla), and MEST+EM | 494MB        | 247MB        |
> | MEST+EM&S                                   | 744MB        | 494MB        |
>
>
> ___
> ### **Q3: Why use a quite low batch size of 64, which challenges the feasibility of this work?**
> I think there is some misunderstanding here.
> The batch size of 64/128 is the standard training batch size for the CIFAR dataset [R1][R2]. We have tried both 64 and 128, where the results show that using a batch size of 64 provides similar or slightly higher accuracy than using a batch size of 128. And it is more friendly for edge devices. Thus, we choose 64 as the batch size.
> All the accuracy results on the CIFAR-10/100 dataset reported in this paper are obtained by using a training batch size of 64 as we mentioned in line 313 and appendix G. So, using a batch size of 64 is acceptable.
>
> [R1] K. He and et.al., Identity Mappings in Deep Residual Networks, ECCV2016
>
> [R2] G. Huang and et.al., Densely Connected Convolutional Networks, CVPR2018
>
>
> ___
> ### **Q4: What is the criteria that limits the practicality/compatibility of these approaches?**
> (1) For the “practicality” in line 306, since the size of the ImageNet dataset itself is about 150GB and a ResNet50 requires more than 1 day to train on a 8 x 2080Ti GPU server, it is impractical to be trained on current edge devices such as mobile phones.
>
> (2) For the “compatibility” in line 359, we intended to say that sparse training works such as the SNIP and GraSP are harder and inefficient to be conducted on edge devices. This is because SNIP and GraSP require to compute the forward and backward propagation of a dense model to find a desired sparse structure. We will give a clearer explanation in the revision to replace the wording “not compatible”. Thank you for pointing it out. We also provide more detailed explanations in the answer to the next question (**Q5**).
>
>
> ___
> ### **Q5: What is the threshold of memory footprint? Is MEST small enough to fit in mobile devices?**
>
> The threshold of memory footprint depends on the device being used.
> Whether a method is small enough depends on (1) whether the dataset and model can be accommodated by a mobile device; (2) whether the free space of device memory (RAM) is sufficient for the required training memory footprint.
>
> And the answer to the second question is yes, MEST is small enough for small/medium dataset training such as CIFAR10/100 on general mobile devices. The current mobile devices generally have memory in GB levels. For example, current general mobile devices such as Samsung Galaxy A20s, Google Pixel 3, and Samsung S20 have 2GB or more memory.
>
> However, unlike the training on a high-end GPU cluster where all the memory can be reserved for training, the memory on mobile devices will also be partially occupied by the operating system and other backend applications. This puts an even greater strain on the memory of mobile devices. Thus, having a smaller memory footprint will always benefit the training on mobile devices.
>
> We use ResNet32 and VGG19 as examples. Assume 32 bits weights, 8 bits indices are used with a batch size of 64.
>
> For ResNet32, the dense model and the methods that involve dense computations require a 462MB memory footprint for model size, while our sparse model with unstructured sparsity requires 46MB (69MB) and 23MB (46MB) under 90% and 95% sparsity for MEST+EM ( MEST+EM&S), respectively.
> For VGG19, a significant reduction can be obtained, where the dense model requires 4964MB memory footprint, while our sparse model with unstructured sparsity requires 494MB (744MB) and 247MB (494MB) under 90% and 95% sparsity for MEST+EM (MEST+EM&S), respectively.
>
> MEST successfully reduces the memory footprint (to less than 100M and 800M for ResNet32 and VGG19) while maintaining a similar or higher accuracy than prior works, while the reference methods either require a dense memory footprint (LT, SNIP, GraSP, RigL) or suffer a severer accuracy degradation.
>
> We will make it clear in the revision.
>
>
> ___
> ### **Q6: Choose 90% sparsity in figure 4c for acceleration results seems unfair for 95% sparsity because 90% has better accuracy compared to baselines while 95% has similar results. Need to show the worst case.**
> Thank you for your suggestion. We think showing the results under 90% sparsity in the main paper could be a good choice.
> We have the following considerations:
> (1) 90% is the sparsity that the reference works begin to have a noticeable acceleration while having a relatively decent accuracy (as shown in Figure 2).
> (2) Under the same sparsity scheme, we have an even larger accuracy advantage over reference works under 95% sparsity compared to the 90% sparsity (i.e., 0.83% v.s. 0.3%, see Table 1).
> (3) Including the reference works, under 95% sparsity, the highest accuracy achieved among all results (i.e., our MEST+EM&S) has around 2.5% accuracy degradation compared to the original model (94.88%). This is a non-trivial degradation on the CIFAR10 dataset, which may make it less representative compared to the results under 90% sparsity.
>
> But we agree that adding a comparison result for 95% sparsity will be helpful. We will add it in the revision.

---

> ### Author Response · Authors · 2021-08-10
> **Author responses to reviewer wx9R (Part 2/2)**
>
> ### **Part 2**
> ___
> ### **Q7: Author claims FLOPs don’t account for execution overheads from irregular sparsity, but still use it in Table 2. Why?**
> We agree that FLOPs cannot precisely represent the training speed for irregular sparsity. We still use it in Table 2 for two reasons.
> (1) In order to make a quantitative comparison to the state-of-the-art works, such as SET[12], DSR[13], RigL[14], etc, we use FLOPs as the same metric to demonstrate the benefit of our proposal.
> (2) Due to hardware inefficiency, reduction of FLOPs for irregular sparsity cannot proportionally translate to the speedup in real devices. Nonetheless, FLOPs serve as an upper bound if new and efficient hardware can be designed to accelerate irregular sparse models. Therefore, it is still informative in Table 2 to demonstrate the FLOPs as theoretical speedup for sparse models. Under such an assumption, we can see that our methods achieve higher accuracy than other works with the same or lower training FLOPs.
>
>
> ___
> ### **Q8: Line 349, author claims MEST+EM can improve channel sparsity. Does the author have any results? Does this optimization bring channel sparsity within Figure 4 (c)’s range?**
> Thank you for the good question. Yes, we have this result. We didn’t show the accuracy of channel sparsity using MEST (vanilla, i.e., without EM) due to the space limitation. The accuracy of using MEST+EM and MEST+EM&S can be found in Figure 4 (a) and (b).
> We show the examples of the accuracy comparison of channel sparsity on ResNet32 on CIFAR100 and CIFAR10 here.
> We can see that our MEST+EM and MEST+EM&S can effectively improve the accuracy, especially under higher sparsity.
>
> **Table A. Accuracy comparison on ResNet32 with channel sparsity on CIFAR100.**
>
> >| sparsity       | dense | 60%   | 70%   | 80%   | 90%   | 95%   | 98%   |
> |----------------|-------|-------|-------|-------|-------|-------|-------|
> | MEST (vanilla) | 74.94 | 70.25 | 68.93 | 67.03 | 63.17 | 58.59 | 51.85 |
> | MEST+EM        | 74.94 | 70.81 | 69.23 | 67.46 | 63.42 | 59.39 | 52.38 |
> | MEST+EM&S      | 74.94 | 70.97 | 69.49 | 67.88 | 64.33 | 60.31 | 54.08 |
>
>
> **Table B. Accuracy comparison on ResNet32 with channel sparsity on CIFAR10.**
>
> >| sparsity       | dense | 60%   | 70%   | 80%   | 90%   | 95%   | 98%   |
> |----------------|-------|-------|-------|-------|-------|-------|-------|
> | MEST (vanilla) | 94.88 | 92.77 | 92.15 | 91.16 | 89.24 | 86.20 | 82.11 |
> | MEST+EM        | 94.88 | 93.12 | 92.72 | 91.64 | 90.31 | 87.49 | 83.03 |
> | MEST+EM&S      | 94.88 | 93.15 | 92.82 | 91.87 | 90.47 | 87.92 | 83.16 |
>
>
> As shown in Table A and Table B, the results corresponding to Figure 4(c) are:
>
> * 63.42% accuracy and 3.1X acceleration for MEST+EM,
>
> * 64.33% accuracy and 2.7X acceleration for MEST+EM&S.
>
> This also indicates that the investigation on sparsity schemes is very necessary for the sparse training acceleration area, because the pattern-based sparsity can provide a comparable acceleration while maintaining a higher accuracy.
>
> * 69.3% accuracy and 2.3X acceleration for MEST+EM,
>
> * 70.8% accuracy and 1.9X acceleration for MEST+EM&S.
>
> We will add another figure for this in the revision.
>
>
> ___
> ### **Q9: reworded "the best-performant MEST" to "the best-performant MEST for the metric"**
> Sorry for the confusion. In Figure 4 (c), when comparing to SNIP, GraSP, SET, and DSR, the best-performant MEST refers to the 90% sparse model labeled “Pattern” and “MEST+EM&S+DE” (the top-right point in purple). It improves the tradeoff among accuracy, training speed, and memory footprint in all 3 aspects. We will make it clear in the revision.
>
>
> ___
> ### **Q10: The concerns mentioned in the limitation and social impact part.**
> Please refer to the answers to questions **Q2** and **Q5**. We will provide clearer explanations to address the concerns in the revision.
> We would like to thank you again for your valuable time and constructive feedback.

---

> ### Comment · Reviewer_wx9R · 2021-08-24
> **Rebuttal Response**
>
> I appreciate the authors providing detailed and on-point responses to all my concerns. I have updated my final rating accordingly.

---

> > ### Author Response · Authors · 2021-08-24
> > **Author Response**
> >
> > Thank you very much for your response and for raising the score. Your valuable comments are constructive, indeed helping us to improve the quality of our submission. We will reflect them in the revision.

---

### Official Review · Reviewer_Knx5 · 2021-07-17

**Rating:** 8
**Confidence:** 4

**Summary:**

This paper proposes the memory-economic sparse training (MEST) framework that targets efficient sparse training on edge devices. The author proposes an elastic mutation (EM) and a soft memory bound algorithm for sparse training. Their training methods can keep the entire training process sparse and avoiding involve dense forward/backward computation. This paper also investigates the impact of different sparsity schemes on sparse training performance in accuracy and speed acceleration. Besides, the paper also introduces data-efficient training in sparse training scenes. Compared to previous works, this work is more hardware friendly and can achieve similar even higher accuracy. These proposed methods are more hardware-friendly and achieve higher accuracy. The proposed framework is promising to be implemented for real-world sparse training acceleration. This paper can be impactful for both the community of academia and industry.

**Limitations And Societal Impact:**

Yes.

**Main Review:**

Originality and Significance:
Different from previous DST works that focus more on the accuracy, this paper investigates the sparse training problem from the perspective of memory economic. There is a trend that sparse training works tend to utilize dense model computation more or less, (e.g., obtain dense gradients and select a new subnetwork based on that). This paper avoids that and only uses sparse computation through the entire sparse training process. This is critical when considering the realistic implementation, especially on edge devices. The proposed EM and soft memory-bound methods are not complicated, but the results look impressively good.

I think one important contribution of this work is that they investigate the performance of different sparsity schemes in adynamic sparse training scenes. The difficulty of unstructured sparsity to achieve actual acceleration is a well-known challenge. There is a long-standing gap between theoretical research and actual implementation of sparse training acceleration, because the research mainly focuses on unstructured sparsity. This work may provide a new direction for the sparse training community to investigate more sparsity schemes that can achieve both acceleration and high accuracy. I always expect to see some works like this one trying to close this gap.

Another contribution of this paper is to introduce data-efficient training, which only uses a partial training dataset, into the dynamic sparse training process, while the accuracy can also be maintained. The data-efficient training itself is not a new concept, but it is novel to be combined with dynamic sparse training and different sparsity schemes. The author also provides a thorough study on how their proposed EM method, algorithms, and sparsity schemes will affect the performance of data-efficient training such as final accuracy and number of unforgettable examples, etc., which is interesting.

Quality and Clarity:
The paper contains quite a lot of content within a 9 pages paper. But overall, the paper is well written and easy to follow. The motivation and contribution of the paper are clearly stated. The experiments are able to support their claims and conclusions. And the superior results show the effectiveness of the proposed methods.

Overall Comments:
Overall, this is a good paper. It investigates sparse training from different angles compared to previous works. It is valuable that this paper takes into account many practical problems in the design, such as memory bound and efficiency of sparse computations, rather than based on pure theoretical analysis. These are critical in real implementation but easy to be ignored in research.
Their proposed methods seem more practical. That is why I think this paper can be impactful for both industry and academia.

Improvements:
For the Data-efficient Sparse Training on the Edge Section (Section 4.2), it would be better if you can draw a figure to show the flow of your two-phase data-efficient sparse training process.

There is a typo in the caption of figure 3. You mentioned the threshold is 1 and 2, where the figure shows th=0 and 1.

Questions：
How do you determine the frequency of the mutation? What is the impact if you have a higher or lower mutation frequency?

I am curious about how to mutate the model topology if you avoid storing a dense model on the edge devices?


**Time Spent Reviewing:**

2

---

> ### Author Response · Authors · 2021-08-10
> **Author responses to reviewer Knx5**
>
> **We would like to thank you for your valuable feedback and comments.**
> ___
> ### **Q1: Typo in Figure 3.**
> Thank you for pointing it out. It is a typo. You are correct, the threshold should be 0 and 1. We will modify it in the revision.
>
> ___
> ### **Q2: How to determine the frequency of the mutation?**
> In our experiments, we searched for the best-suited mutation frequency heuristically. We found that the mutation frequency cannot be too high or too low. The possible reason is that if a very low frequency is used, then the overall search space of finding desirable sparse model structures will not be sufficient. If a very high frequency is used, then the new sparse structure will not be trained enough. This will compromise the quality of the weight importance estimation for future mutation.
>
> Our results show that the best-suited mutation interval is around 5 epochs on CIFAR10/100 and 2 epochs on ImageNet. This also strengthens our conjecture above. Since the ImageNet is harder than CIFAR10/100, it requires a larger search space, and the number of iterations in each training epoch on ImageNet is much larger than on CIFAR10/100. Putting all together, the best-suited mutation frequency for ImageNet is higher than for CIFAR10/100. We will add this in the revision.
>
> ___
> ### **Q3: How to mutate while keeping it sparse?**
> Thank you for the good question.
> On the edge devices, we store the sparse weights using compressed sparse row (CSR) format (as mentioned in Appendix B). Indices are used to indicate the location of the stored non-zero weights.
> During our mutation process, for the weights that will be removed, we first reset their values to zero. Then, directly replace the indices of the mutated weights with new random indices that do not yet exist. In this way, we can keep the memory footprint for weight storage unchanged all the time.

---

> ### Author Response · Authors · 2021-08-29
> **Author response**
>
> We would like to thank you again for your valuable time and comments. We will fix the typo in Figure 3 and add a discussion about mutation frequency. Thank you for your appreciation of our work!

---

### Official Review · Reviewer_HrFw · 2021-07-18

**Rating:** 7
**Confidence:** 4

**Summary:**

This paper proposes several techniques to boost DNNs' training efficiency on edge devices from the mutation rate schedule and data efficiency perspective. It's good to see the improvements in the training speed and memory footprint measured on real mobile devices, which may practically help the community.

**Limitations And Societal Impact:**

The limitations are discussed in the main review section.

**Main Review:**

## Originality and significance
The proposed techniques are evaluated on real mobile devices with non-trivial improvements in training speed and memory footprint, which is appreciated in DNN acceleration works and may positively help the community. However, most of the techniques are built on top of previous works with marginal novelty, which is my main concern for this paper. For example, considering both weight magnitude and gradient as the pruning metric is widely adopted in previous pruning works. In addition, the proposed EM/EM&S is more like engineering tricks built on top of existing sparse training works which lack motivating analysis to support their general effectiveness, so is the two-phase data efficient training. Therefore, the novelty of this work needs to be better justified.

## Quality and clarity
The paper conduct extensive experiments to validate the effectiveness of the proposed methods, which is solid. However, the following concerns may influence the quality of this paper:

(1) The writing can be better organized. As the data efficient training has been discussed in many previous works like E2Train (Y. Wang, NeurIPS'19) as well as [20][21][22][23] cited by this paper, the two-phase training is marginally different from previous solutions which can be simplified. Instead, Sec. 3.2 is too concise and many details in the appendix are necessary for demonstrating the proposed techniques. It may be better to re-balance the length of each technique in this paper.

(2) Although the evaluation shows impressive improvements, the motivation and rationale behind each technique are not well-analyzed, especially in Sec. 3.2. This may limit the insights provided by this paper.

(3) The paper mainly considers ResNet-32/VGG-19 for experiments on CIFAR-10/100 and it may strength the evaluation to evaluate the proposed technique on (a) MobileNetV2, as compact models may suffer more from high sparsity, and (b) deeper ResNet like ResNet-110, as deeper networks may suffer more from gradient explosion/vanishing problems under compression techniques.

(4) To confirm some details:

(a) According to algorithm 1, will the proposed techniques be applied on gradients and activations as well?

(b) Considering the unforgettable examples may be less on ImageNet, how many data will be discarded by MEST + EM&S + DE on ImageNet in Table 2? It seems w/ and w/o DE makes trivial difference?

I'm willing to raise my scores if the above concerns are properly addressed.

**Time Spent Reviewing:**

1.5

---

> ### Author Response · Authors · 2021-08-10
> **Author responses to reviewer HrFw**
>
>
> **Thanks for your valuable feedback and comments. We will definitely reflect them in the paper to improve our work. We clarify the concern about significance of our work, explain rationales behind our techniques, and complement evaluations with additional results as per reviewer’s feedback.**
>
> ___
>
> ### **Q1: MEST is built on top of previous works with marginal novelty.**
>
> Thanks for your acknowledgment of our systematic evaluation on real mobile devices, non-trivial improvements in training speed and memory footprint, and positive impacts to the DNN acceleration community.
>
> In this work, we are attempting to develop novel edge training techniques that are not only accurate, but also efficient in terms of training acceleration and memory footprint to facilitate execution on edge mobile devices, and have the potential to benefit subsequent inference speedup. This is a challenging task and prior arts mainly contribute at the algorithm level.
>
> Different from prior arts and bearing in mind the ultimate goal of edge training on real mobile devices, this work proposes a series of technical contributions such as: incorporating both weight and gradient as the importance of weight, sparse training algorithm improvements by EM and EM&S, and data efficiency. Although these techniques seem to be straightforward or familiar to some other domains as per the reviewer’s comments, it is no doubt that they are effective, first attempts for sparse training on the edge devices. Specifically, these techniques enable us to leverage model sparsity to fully exploit characteristics of real mobile devices.
>
> Additionally, it is the first systematic investigation about the impacts of sparsity schemes on accuracy, memory footprint, and the actual acceleration rate of sparse training. These are very important in practical sparse training but missing in prior arts. We try to draw the community’s attention to these aspects and facilitate the study of practical sparse training acceleration on edge devices.
>
> Last but not least, data-efficient training although has been studied in prior works, it has never been investigated for sparse training. This is not a trivial combination. Because data efficiency performs differently in dynamic sparse training scenes than in dense training. All factors including sparsity scheme, sparsity ratio, number of epochs of each training phase, etc., will affect the number of unforgettable examples, final accuracy, and overall training cost on real devices. In this paper, we thoroughly investigate the characteristics of data efficiency for sparse training in Fig. 3 and more in Appendix F. Additionally, as mentioned in lines 384~391 and in Appendix J, we comment that there is still a lot of room for exploration in this area. We believe our work will be impactful and motivate future works in this direction.
>
> ___
> ### **Q2: Re-balance the length of each technique. Sec 3.2 too concise. Motivation and rationale behind each technique.**
>
> Thanks for the suggestion. We will make the points more explicit in the paper.
>
> About motivation and rationale, the motivation (objective) of the overall MEST is in lines 173\~174. The rationale of the “mutation” operation is explained in lines 175\~177.
> The rationale of EM is in lines 182\~183. The thoughts about EM are: (1) a larger mutation ratio will provide a larger search space during the dynamic sparse training process; and (2) the dramatic structural change of the network may compromise the training convergence. Thus, we propose our EM method to gradually reduce the mutation ratio during the dynamic sparse training process, which can keep a large search space and also help stabilize the model structure, as mentioned in lines 182\~183. We will add the above two thoughts in the paper.
> The motivation of EM&S is in lines 191\~193. Further explanations are as follows: Different from EM that the less important weights will always be removed, our EM&S allows the newly grown weights to be added to the existing weights and then trained, then the unimportant weights will be selected from all weights including the newly grown weights. This can avoid forcing the existing weights in the model to be removed if they are more important than newly grown weights. This can be considered as adding an ‘undo’ mechanism to the mutation process. We will incorporate these into the paper as well.
>
> About the length of Sec 3.2, we also have Appendices D and E to explain our techniques. Besides adding the above mentioned details, we will move key points in Appendices D and E to the paper.
>
> ___
> ### **Q3: Needs to evaluate with MobileNet-v2 and ResNet-110**
>
> We choose ResNet-32 and VGG19 in the paper for a fair comparison, since they are used as the benchmark networks in many representative sparse training works. And thank you for your suggestion. We agree adding more networks will help readers for better understanding. Below are some results on MobileNetv2 and ResNet110, and we will add them in the paper.
>
> Table A shows the accuracy achieved by our MEST+EM&S method on CIFAR10 under different sparsity ratios, while Table B shows the overall training FLOPs (1e^12) and number of model parameters (M).
> You are right, the MobileNetv2 is more sensitive to sparsity compared to ResNet32 and ResNet110, as shown in Table A. This is why we can barely see MobileNet models to be used in pruning or sparse training works.
> Actually, the ResNet32 (as well as ResNet20, ResNet110) are the lightweight ResNet version dedicated to CIFAR tasks, while the ResNet18, ResNet34, and ResNet101 are the large versions for the ImageNet. So, as we can see from Table B, the ResNet32 is smaller than MobileNetv2 (1.86M params. v.s. 2.3M params.), while the computation cost of ResNet32 is higher than MobileNetv2 due to the depth-wise separable CONV. Note that the ResNet32 here (and in the paper) is a (2x) widened version, which is consistent with the reference works cited in the paper (all as shown in Table 1 in the main paper). This is the reason that the number of parameters and training cost of ResNet32 is similar to the ResNet110 as shown in Table B.
>
> It is interesting to see that, under 90% sparsity, ResNet32 has a similar accuracy and training FLOPs as the MobileNetv2 under 60% sparsity, while the number of parameters of ResNet32 is 4.8X less than MobileNetv2. For this case, the ResNet32 will be more desired than MobileNetv2.
> Moreover, MobileNetv2 is much deeper (57 CONV layers) than ResNet32, which will require more data movement among memory and cache for reading and writing intermediate results and lead to a higher execution overhead.
> We are grateful for your suggestion here which leads to these meaningful and interesting results and discussions.
>
>
> **Table A. Accuracy comparison on CIFAR10 using MEST+EM&S with unstructured sparsity.**
>
> >| Sparsity    | Dense | 50%   | 60%   | 70%   | 80%   | 90%   |
> |-------------|-------|-------|-------|-------|-------|-------|
> | ResNet32    | 94.88 | 94.41 | 94.05 | 94.14 | 93.7  | **93.27** |
> | MobileNetv2 | 94.08 | 94.06 | **93.32** | 93.05 | 92.38 | 90.61 |
> | ResNet110   | 94.64 | 93.47 | 93.73 | 93.62 | 93.26 | **92.29** |
>
>
> **Table B. Training FLOPs and number of parameters comparison on CIFAR10 using MEST+EM&S with unstructured sparsity.**
>
> >| Train FLOPs(1e12) / Params(M) | Dense       | 50%         | 60%         | 70%         | 80%         | 90%         |
> |-------------------------------|-------------|-------------|-------------|-------------|-------------|-------------|
> | ResNet32                      | 6.38/1.86 | 3.30/0.93 | 2.68/0.74 | 2.07/0.56 | 1.45/0.37 | **0.83/0.19** |
> | MobileNetv2                   | 2.11/2.30 | 1.09/1.15 | **0.88/0.92** | 0.68/0.69 | 0.48/0.46 | 0.28/0.23 |
> | ResNet110                     | 5.74/1.70 | 2.97/0.85 | 2.41/0.68 | 1.86/0.51 | 1.31/0.34 | **0.75/0.17** |
>
> ___
> ### **Q4: Will the proposed techniques be applied to gradients and activation according to algorithm 1?**
>
> The weights and gradients share the same sparsity topology, so the proposed techniques are applied to gradients, but not the activation. Due to the sparsity in weights and gradients, both forward and backward propagation phases in the training can have reduced computations compared to the dense training. Please also refer to Appendix C.1 for details.
>
> ___
> ### **Q5: ImageNet on Table 2, how much data will be discarded by MEST + EM&S + DE?  w/ and w/o DE makes trivial difference?**
>
> In Table 2, since we intend to maintain the final accuracy, we remove 10%\~13% of training examples and the final accuracy is indistinguishable to training the whole dataset. This provides an extra training speedup on ImageNet.
> Since Phase-one requires a full dataset training to statistic the unforgettable examples, the training cost reduction comes from Phase-two.
> We also want to mention that both removing more training examples and increasing the model sparsity can make a tradeoff between training speed and accuracy. If removing more training examples is desirable, then one can use a less sparse model to maintain the final accuracy.
> How to balance the model sparsity and the reduction of training examples is an interesting topic for future work as we discussed in the paper (we also show an ablation study in Appendix J).

---

> > ### Comment · Reviewer_HrFw · 2021-08-24
> > **Follow up**
> >
> > Thanks for the detailed response and extensive experiments! After reading the author feedback, I agree that the proposed techniques are pioneering attempts of extensively investigating sparse training on edge devices, which will greatly benefit the community. And the authors provide solid experiments and explanations for addressing my concerns. I will raise my score to a clear accept.

---

> > > ### Author Response · Authors · 2021-08-25
> > > **Author Response**
> > >
> > > Thank you for your appreciation of our work and for raising the score. Your comments are very constructive, e.g., re-balancing the sections, adding results for more networks, and enriching our analyses and discussions, which make our paper stronger. We will address them carefully in the revision. And thank you again for your valuable time.

---

### Decision · Program_Chairs · 2021-09-27

**Decision:**

Accept (Spotlight)

**Comment:**

This paper proposes the memory-economic sparse training (MEST) framework for efficient sparse training on edge devices. The author proposes an elastic mutation (EM) and a soft memory bound algorithm for sparse training. Their training methods can keep the entire training process sparse and avoiding involve dense forward/backward computation. This paper also investigates the impact of different sparsity schemes on sparse training performance in accuracy and speed acceleration. Besides, the method also introduces data-efficient training in sparse training scenarios.

The rebuttal has resolved most reviewers' concerns. All reviewers are quite possible on this paper.